# Coupling the Canadian Terrestrial Ecosystem Model (CTEM v. 2.0) to Environment and Climate Change Canada's greenhouse gas forecast model (v.107-glb)

Bakr Badawy[1,*], Saroja Polavarapu[1], Dylan B. A. Jones[2], Feng Deng[2], Michael Neish[1], Joe R. Melton[3], Ray Nassar[1], and Vivek K. Arora[3]

[1]Climate Research Division, Environment and Climate Change Canada, Toronto, Canada
[2]Department of Physics, University of Toronto, Toronto, Canada
[3]Climate Research Division, Environment and Climate Change Canada, Victoria, Canada
[*]Now at Faculty of Environment, University of Waterloo, Canada

*Correspondence to:* Bakr Badawy(bbadawy@uwaterloo.ca)

**Abstract.** The Canadian Land Surface Scheme and the Canadian Terrestrial Ecosystem Model (CLASS-CTEM) together form the land surface component in the family of Canadian Earth System Models (CanESM). Here, CLASS-CTEM is coupled to Environment and Climate Change Canada (ECCC)'s weather and greenhouse gas forecast model (GEM-MACH-GHG) to consistently model atmosphere-land exchange of $CO_2$. The coupling between the land and the atmospheric transport model

ensures consistency between meteorological forcing of $CO_2$ fluxes and $CO_2$ transport. The procedure used to spin up carbon pools for CLASS-CTEM for multi-decadal simulations needed to be significantly altered to deal with the limited availability of consistent meteorological information from a constantly changing operational environment in the GEM-MACH-GHG model. Despite the limitations in the spin up procedure, the simulated fluxes obtained by driving the CLASS-CTEM model with meteorological forcing from GEM-MACH-GHG were comparable to those obtained from CLASS-CTEM when it is driven with

standard meteorological forcing (CRU-NCEP). This is due to the similarity of the two meteorological datasets in terms of temperature and radiation. However notable discrepancies in the seasonal variation and spatial patterns of precipitation estimates, especially in the tropics, were reflected in the estimated carbon fluxes, as they significantly affected the magnitude of the vegetation productivity and, to a lesser extent, the seasonal variations in carbon fluxes. Nevertheless, the simulated fluxes based on the meteorological forcing from the GEM-MACH-GHG model are consistent to some extent with other estimates

from bottom-up or top-down approaches. Indeed, when simulated fluxes obtained by driving the CLASS-CTEM model with meteorological data from the GEM-MACH-GHG model are used as prior estimates for an atmospheric $CO_2$ inversion analysis using the adjoint of the GEOS-Chem model, the retrieved $CO_2$ flux estimates are comparable to those obtained from other systems in terms of the global budget and the total flux estimates for the northern extratropical regions, which have good observational coverage. In data poor regions, as expected, differences in the retrieved fluxes due to the prior fluxes become apparent.

Coupling CLASS-CTEM into EC-CAS is considered an important step toward understanding how meteorological uncertainties affect both $CO_2$ flux estimates and modelled atmospheric transport. Ultimately, such an approach will provide more direct

feedback to the CLASS-CTEM developers, and thus help to improve the performance of CLASS-CTEM by identifying the model limitations based on atmospheric constraints.

*Copyright statement.* TEXT

## 1 Introduction

Terrestrial ecosystems play a crucial role in the global climate-carbon system. Therefore, there is a need to better understand terrestrial biospheric processes related to the carbon cycle in order to obtain more reliable projections of their behavior under a changing climate. Given the great heterogeneity of vegetation and soils, the coverage and accuracy of the flux measurements are not sufficient for obtaining large-scale flux estimates with high confidence (Jung et al., 2009; Beer et al., 2010). As a result, considerable efforts have been made to develop terrestrial ecosystem models (TEMs) (whether simple regression or process-

oriented) in order to quantify the magnitude, geographical distribution, and evolution of sources and sinks of carbon at regional and global scales (Potter et al., 1993; McGuire et al., 2001; Sitch et al., 2003; Thornton et al., 2005; Krinner et al., 2005; Reichstein et al., 2005; Badawy et al., 2013; Arora and Boer, 2005; Melton and Arora, 2016). However, systematic errors and uncertainties in the models can result from driving or forcing data (Jung et al., 2007; Clein et al., 2007; Zhao et al., 2006; Garnaud et al., 2014; Dalmonech et al., 2015; Anav et al., 2015; Wei et al., 2014), process formulation (also called model

structure) (Sitch et al., 2015), model parameter specification, and initial conditions (Carvalhais et al., 2008, 2010; Melton et al., 2015; Zhu and Zhuang, 2015), leading to differing estimates of $CO_2$ fluxes from different models (McGuire et al., 2001; Piao et al., 2013; Sitch et al., 2015). Such differences in TEMs are among the main sources of uncertainty in future projections from coupled carbon-climate models (Anav et al., 2013; Friedlingstein et al., 2006; Arora et al., 2013; Friedlingstein et al., 2014). Therefore, there is a need to evaluate the performance of TEMs in order to identify and diagnose their weaknesses and

strengths and ultimately reduce model uncertainties. Indeed, this is the motivation behind TEM multimodel intercomparisons efforts such as the Multi-scale Synthesis and Terrestrial Ecosystem Model Intercomparison Project (MsTMIP) (Huntzinger et al., 2013).

   Inverse models (which relate observed concentrations to fluxes using an atmospheric transport model) are powerful tools to quantify carbon fluxes over large regions (Rödenbeck et al., 2003; Peters et al., 2007; Peylin et al., 2013) and can be used to

evaluate the TEM results. However, inverse models suffer from deficiencies and uncertainties (Peylin et al., 2013) arising from transport errors, choice of observation network, observation uncertainties, and prior flux errors. Inverse models also provide little or no information about the underlying processes responsible for the estimated fluxes. Hence, they cannot be used to understand or predict the future behavior of the carbon cycle. Alternatively, there are carbon cycle data assimilation systems (CCDAS), which couple the strengths of the top-down (inversion) and bottom-up (i.e. TEM) approaches by embedding a TEM

within a comprehensive climate model and using measurements from multiple streams to constrain the TEM (Scholze et al., 2003; Rayner et al., 2005; Koffi et al., 2013). The benefit is that biospheric models can then be validated on the global scale

using atmospheric measurements of $CO_2$ that integrate the $CO_2$ signal at various spatial and temporal scales. In CCDAS, key parameters of a TEM can also be optimized to improve its fit to atmospheric $CO_2$ observations (Scholze et al., 2003; Rayner et al., 2005; Koffi et al., 2013; Kaminski et al., 2013), which can potentially yield greater understanding about underlying processes, and thus can help to improve the model performance. Nevertheless, CCDAS is sometimes challenging and has its

limitations. For example, the optimized fluxes (and parameters) are sensitive to CCDAS configurations (Kaminski et al., 2013) (i.e, atmospheric transport, background fluxes, observational network, processes representations, missing process, etc).

Comprehensive Earth System Models need to include TEMs because the ecosystem responds to a changing climate. However, weather and carbon fluxes are also interconnected so that coupled weather and $CO_2$ prediction models operating on weather or seasonal timescales can also benefit from online TEMs. Previous studies (Lin et al., 2011; Garnaud et al., 2014)

have shown that uncertainties in meteorological forcings (i.e. temperature, specific humidity, shortwave radiation) contribute significantly to uncertainties in the simulated fluxes. In addition, Miller et al. (2015) have shown that several meteorological variables (i.e. temperature, specific humidity, zonal wind, and planetary boundary layer) are correlated and contribute to biases in modelled atmospheric transport. Therefore, inconsistencies may arise when the TEM's meteorological driving data differs from that in the weather model. For example, at a given point in time, a TEM's grid cell might have experienced sunny weather

and thus produced large $CO_2$ uptake whereas the weather model may indicate cloudy conditions and reduced $CO_2$ uptake. An online TEM constrained by the model's weather would have predicted this reduced $CO_2$ uptake. If such inconsistent $CO_2$ predictions are used to constrain inverse models there is a risk of misattributing some of the model-data mismatch to the flux estimate. Therefore, coupling between TEMs and atmospheric transport models (i.e. using the same meteorological variables) is considered an important step toward understanding how meteorological uncertainties impact both $CO_2$ flux estimates and

modelled atmospheric transport. Forecasting systems that integrate land and ocean $CO_2$ fluxes within numerical weather prediction (NWP) models have recently been developed (Agusti-Panareda et al., 2014; Ott et al., 2015) to produce short term predictions of atmospheric $CO_2$.

At Environment and Climate Change Canada (ECCC), a Carbon Assimilation System (EC-CAS) (Polavarapu et al., 2016) is being developed by implementing an ensemble Kalman Filter (EnKF) for greenhouse gas state and flux estimation. EC-CAS

will assimilate satellite and in situ data to generate hindcasts of atmospheric $CO_2$ and estimates of regional fluxes of $CO_2$. EC-CAS uses an ensemble of coupled meteorological and constituent forecasts to directly compute the complex transport error covariances needed for proper flux estimation (Miller et al., 2015). The meteorological model is an adaptation of the operational weather prediction model GEM-MACH (Global Environmental Multi-scale - Modelling Air quality and CHemistry) (Moran et al., 2010; Robichaud and Ménard, 2014; Makar et al., 2015) for greenhouse gasses (called GEM-MACH-GHG hereafter,

see Section 2.2). In addition to an ensemble of meteorological fields, EC-CAS also requires an ensemble of prior fluxes to simulate prior flux uncertainty. With an online ecosystem model, we can directly perturb parameters to get this ensemble. In turn, the assimilation process would provide continual feedback on the ecosystem model, just as in CCDAS. For this reason, we envision an online ecosystem model eventually within EC-CAS. However, we first consider the simpler approach of offline coupling wherein the atmospheric model's meteorology is used to drive the ecosystem model. Thus, a key objective of the work

here is to assess the viability of land surface fluxes of $CO_2$ from the Canadian Terrestrial Ecosystem Model (CTEM) (Melton

and Arora, 2014), coupled to the Canadian Land Surface Scheme (CLASS) (Verseghy, 2012), as a source of a priori biospheric fluxes of $CO_2$ for EC-CAS and other $CO_2$ flux inversion systems. CLASS-CTEM is a process-based TEM which simulates the exchange of carbon, water and energy fluxes between the land surface and the atmosphere. It is similar in level of complexity to other TEMs (such as CASA (Potter et al., 1993) or SiB (Sellers et al., 1996)) which have been used for flux inversions and

5 which have participated in multimodel intercomparisons such as that of Huntzinger et al. (2012). In recent studies (Melton and Arora, 2014; Melton et al., 2015; Melton and Arora, 2016; Badawy et al., 2016), CLASS-CTEM was calibrated based on observation-based climate data from the Climate Research Unit (CRU) (Harris et al., 2014) combined with reanalysis fields from the National Centers for Environmental Prediction (NCEP) (Kalnay et al., 1996). By coupling CLASS-CTEM with the atmospheric model, this work helps to pave the way for the coupled meteorological and ecosystem model within the EnKF

(e.g. see conclusions of Miller et al. (2015)).

Although incorporating CLASS-CTEM within EC-CAS is potentially mutually beneficial, the incorporation of a TEM designed for Earth System Modelling (decadal timescales) into a data assimilation system designed for short timescales (i.e. months to a few years) is not without its challenges. For example, the spin-up of carbon pools to present climate needs to be merged with the switch in climate data from reanalyses to that from the weather forecasting model (e.g. EC-CAS). The chal-

15 lenge is that operational weather forecasting systems are, by definition, constantly changing so that long archives of consistent analyses (i.e. with the same horizontal or vertical resolution or model coordinates or variable, etc.) are not available (see also Agusti-Panareda et al. (2017) for example), contrary to the case of reanalyses (e.g. ERA-Interim (Dee et al., 2011) or MERRA (Rienecker et al., 2011)). Given that the spin-up procedure is known to impact TEM predictions (Wutzler and Reichstein, 2007; Carvalhais et al., 2008, 2010), how will this affect the use of CLASS-CTEM in the flux estimation context? In addition, the

20 environmental drivers of TEMs also impact their results, so will the change in the climate forcing of CLASS-CTEM negatively impact its predictions on these short "climate timescales"? Garnaud et al. (2014) show that carbon pools and fluxes from CLASS-CTEM are sensitive to climate datasets for the case of a limited area domain (North America). On the other hand, large changes in fluxes are not necessarily detectable by observing systems such as Greenhouse Gases Observing SATellite (GOSAT) (Ott et al., 2015) so such differences may not be perceptible in flux inversion results. Finally, given that there is

25 already a well-documented sensitivity to prior flux estimates in data sparse regions (Gurney et al., 2004; Peylin et al., 2013), do such deficiencies in spin-up procedure and environmental drivers matter? In other words, despite the unavoidable imperfections in coupling a TEM from an Earth System Model to a Carbon Assimilation System focussed on short climate timescales, will the flux inversion results obtained using CTEM fall within the range of uncertainty encompassed by an ensemble of recognized flux inversion systems? The goal of this work is to answer these questions.

We begin in Section 2 with a description of the various models and datasets involved in this study, followed by the experimental design (Section 3). In order to interpret differences in fluxes resulting from the change in meteorological forcing, we first compare the quality of the meteorological inputs from GEM-MACH-GHG against the standard climate forcing (CRU-NCEP) that was used to drive CLASS-CTEM, as well as against independent sources of data (Section 4.1). Then, we examine the sensitivity of the simulated carbon fluxes to the change in meteorological forcing to determine whether biases in the simulated

carbon fluxes can be attributed to biases in the meteorological variables (Section 4.2). The simulated fluxes are assessed both

directly as well as indirectly through their impact on $CO_2$ concentrations. Finally, in Section 4.3, the a priori fluxes from CTEM are used in a flux inversion system and the results are analyzed in terms of the seasonal cycle and annual totals of the optimized fluxes and the a posteriori $CO_2$ concentrations. The conclusions are presented in Section 5.

## 2  Models and Data

Before presenting the experimental design, we first introduce the TEM and the coupled meteorological and tracer transport model to which the TEM will be coupled. Then, the validation datasets used to assess the various sources of climate forcing are described, followed by the experimental methodology.

### 2.1  CLASS-CTEM

The coupled CLASS-CTEM model used here is based on CLASS v3.6 (Verseghy, 2012) and an updated version of CTEM
v1.2 (Melton and Arora, 2014) and runs globally on a Gaussian 128×64 grid that corresponds to $\sim 2.8° \times 2.8°$ grid spacing. CLASS calculates the biophysical exchange of energy and water fluxes between the land surface (soil, snow, and vegetation canopy) and the atmosphere. The model includes three soil layers, which extend to a total depth of 4.1 m, and one vegetation canopy and one snow layer. The model solves for the energy and hydrological balances at each grid cell using a half-hourly time step. The land surface of each grid cell is divided into four subareas: bare soil, vegetation, snow over bare soil and snow with
vegetation. The vegetation within a grid cell, in CLASS, can be composed of 4 PFTs (Plant Functional Types): needleleaf trees, broadleaf trees, crops and grasses. For each PFT, prescribed physiological characteristics, such as albedo, annual maximum and minimum leaf area index (LAI), vegetation height, canopy mass, and rooting depth have to be specified. When coupled to CTEM, these structural vegetation attributes are dynamically simulated by CTEM with a daily time step and then passed to CLASS.

CTEM is a process-based terrestrial biosphere model that grows vegetation from bare ground and simulates the main processes governing carbon fluxes between the land biosphere and atmosphere. The model is parametrized and designed to simulate land-atmosphere exchanges of carbon through photosynthesis, ecosystem respiration (sum of autotrophic and heterotrophic respiration), phenology, turnover, mortality, allocation, fire and land use change (Arora, 2003; Arora and Boer, 2005; Melton and Arora, 2016). The model is represented by three living vegetation pools (leaves, stems, and roots) and two dead carbon
pools (soil organic matter and litter). The terrestrial ecosystem processes are calculated for nine PFTs: Needleleaf evergreen, Needleleaf deciduous, broadleaf evergreen, broadleaf cold deciduous, broadleaf drought/dry deciduous, crops ($C_3$ and $C_4$) and grasses ($C_3$ and $C_4$). When coupled, CTEM provides time-varying vegetation structure attributes to CLASS and the calculated variables for the nine PFTs are averaged (weighted by the fractional coverage of each PFT) to obtain the four PFTs in CLASS that share similar functionality.

Within CTEM, photosynthesis and leaf respiration sub-modules operate on a half-hourly time step as in CLASS in order to model the effect of the $CO_2$ concentration on stomatal conductance. Other terrestrial ecosystem processes, including stem, root, and heterotrophic respiration are modelled at a daily time step. Recently, Badawy et al. (2016) modified CTEM to add the

capability to simulate all respiratory fluxes at the same time step as CLASS (i.e. half-hourly) in order to model their diurnal variation caused by subdiurnal signals in the driving climate data. The current version of CTEM does not include the nitrogen cycle and its interactions with carbon cycle. Nevertheless, the model constrains the response of terrestrial photosynthesis to elevated $CO_2$ via an empirical formulation based on experimental plant growth studies (Arora et al., 2009). The model structure

and its parametrizations are documented in Arora (2003), Arora and Boer (2005), and Melton and Arora (2016), in which a comprehensive description of model subroutines is provided.

Besides the meteorological inputs (shortwave and longwave downward radiation, air temperature, precipitation, specific humidity, surface pressure, wind speed (see Section 2.4)), the model requires data on soil texture (i.e. percentage of sand and clay for the three soil layers), fractional vegetation coverage for each PFT, organic matter content, permeable soil depth, and

10 atmospheric $CO_2$. The soil texture information is based on Zobler (1986). The vegetation fractional coverage for the nine PFTs in CTEM are adapted from Arora and Boer (2010) but using the HYDE v3.1 data set for crop area (Hurtt et al., 2011) to reconstruct the historical land cover. The model uses inputs of annual mean atmospheric $CO_2$ concentrations, which are based on phase 5 of the Coupled model Intercomparison Project (CMIP5) (Meinshausen et al., 2011).

## 2.2 GEM-MACH-GHG

GEM-MACH is based on the dynamics and physics of the Global Environmental Multiscale (GEM) model (Côté et al., 1998a; Girard et al., 2013) at the Canadian Meteorological Centre (CMC). GEM is used for operational weather forecasting in both global and regional (North America) domains, whereas GEM-MACH includes an online chemical model that is fully integrated into the meteorological model to provide air quality forecasts over North America. GEM-MACH-GHG (v.107-glb) (Polavarapu et al., 2016) is a variant of GEM-MACH that removes the reactive chemistry and replaces it with climate-chemistry (e.g. OH

climatology). In addition, a number of modifications to GEM-MACH were made, including the implementation of a mass conservation scheme, and modifying the vertical mixing in the boundary layer. A horizontal resolution of $0.9°$ ($400 \times 200$ grid points), and a time step of 30 minutes are used.

In this study, the meteorological fields required to drive CLASS-CTEM are produced from GEM-MACH-GHG (v.107-glb) following the same approach as in Polavarapu et al. (2016) for the 2009-2010 period. Prior to 22 June 2009, the operational

analyses were produced using a model with a lid at 10 hPa. Since that date, the operational model has used a much higher lid of 0.1 hPa and since the period of interest for greenhouse gas simulations commences with the launch of the Greenhouse Gases Observing Satellite (GOSAT) (Kuze et al., 2009; Yokota et al., 2009) in 2009, GEM-MACH-GHG uses the more recent model configuration. As a result, it is difficult to make use of CMC analyses prior to 22 June 2009. Thus, early in 2009, these analyses were supplemented by CMC archives of the "parallel run" (the system during its testing phase) and a preliminary

run. Given that GEM-MACH-GHG was under development during this study, only a few years were simulated (2009-2010). There will always be an unsatisfactory length of analyses available for a TEM spin up period whenever operational weather forecast system is involved. Moreover, greenhouse gas assimilation systems are constrained (by time, computational expense and the observing system) and thus often focus on a few years of study at one time (e.g Deng et al. (2014, 2016)). Thus, the challenge is to merge this small dataset into the spin-up procedure used for the TEM. As we shall see, despite this considerable

challenge, the resulting net fluxes are still comparable to other observation-based estimates and model simulations. The meteorological fields are initialized at the start of each 24h cycle with archived analyses from the CMC which were produced by the previously operational four-dimensional variational (4D-Var) data assimilation system (Charron et al., 2012), interpolated to GEM-MACH-GHG's $0.9°$ resolution. The 24-hour forecasts of shortwave and longwave radiation, surface temperature, wind speed, surface pressure, total precipitation, and specific humidity, were generated every 30 minutes, and then interpolated to the CLASS-CTEM grid.

## 2.3 GEOS-Chem

Previous inversion studies show that optimized fluxes are sensitive to prior fluxes particularly for regions that are poorly constrained by atmospheric observations such as the tropics (Peylin et al., 2013). In order to assess the quality of NEE from CTEM-GEM in comparison to other flux estimates, it is necessary to perform some inversion studies. Ideally, such inversions would be conducted with GEM-MACH-GHG but since the assimilation capability of EC-CAS is still under development, an alternative inversion system based on the GEOS-Chem model (http://geos-chem.org) is used. The GEOS-Chem model has often been used to simulate atmospheric $CO_2$ (e.g. Suntharalingam et al. (2004); Nassar et al. (2010)). This model is a global 3-D chemical transport model driven by assimilated meteorology from the Goddard Earth Observing System (GEOS-5) of the NASA Global Modelling and Assimilation Office (GMAO). Nassar et al. (2010) described an update of the atmospheric $CO_2$ simulation in GEOS-Chem. In this study, the model has a horizontal resolution of $4° \times 5°$, with 47 vertical layers from the surface to 0.01 hPa. The assimilation system is a 4D-Var data assimilation system in which a set of scaling factors is optimized to adjust the fluxes in each model grid box to better reproduce the observations over a given time period. In the 4D-Var system, the adjoint of the GEOS-Chem model is used to optimize the fluxes. Details of the GEOS-Chem adjoint model are given in Henze et al. (2007) and a description of its application for inverse modelling of atmospheric $CO_2$ is provided in Deng et al. (2014, 2016).

## 2.4 CRU-NCEP

The observation-based $0.5°$ monthly climatology from the Climate Research Unit (CRU, version TS3.2) (Harris et al., 2014) and the $\sim 2.5°$, 6-hourly reanalysis fields from the National Centers for Environmental Prediction (NCEP) (Kalnay et al., 1996) were combined to produce the CRU-NCEP global climate data set (Viovy, 2016) that has been described in Wei et al. (2014). The CRU-NCEP dataset provides globally gridded ($0.5° \times 0.5°$) 6-hourly time-varying climatology products that covers the period 1901-2014. The input data from CRU-NCEP includes shortwave and longwave radiation, surface temperature, wind speed, surface pressure, total precipitation, and specific humidity. These climate data were interpolated to the CLASS-CTEM's grid and disaggregated to a half-hourly time step as described in Arora and Boer (2005), and Melton and Arora (2014).

## 2.5 Other Datasets

To evaluate the quality of the GEM driving data, the forecasted fields of shortwave radiation, temperature, and precipitation for 2009 and 2010 are compared with CRU-NCEP, and both are evaluated against the CRU dataset and the ERA-Interim reanalysis (hereafter called ERAI) of the European Centre for Medium-Range Weather Forecasts (ECMWF) (Berrisford et al., 2011; Dee et al., 2011). The $2.5°$ monthly ERAI data is available at the ECMWF data server. Comparing to ERAI is done only to get a sense of how well a reputable reanalysis product compares to independent observations. Since the GEM products are only analyses they are not expected to perform as well as reanalysis products like ERAI. Thus, if reanalyses also have difficulty in matching observations, this provides context or bounds for the kind of agreement we can expect from our analyses.

To assess the impact of using alternative driving data on the simulated fluxes, the CLASS-CTEM fluxes obtained with GEM and CRU-NCEP meteorology are compared and evaluated against independent observation-based flux estimates and other model results. For example, the simulated GPP was compared with the upscaled GPP from FLUXCOM (Jung et al., 2017). The FLUXCOM GPP data is based on machine learning methods that integrate in situ flux measurements, satellite-based vegetation indices, and meteorological data (Tramontana et al., 2016; Jung et al., 2017). In this study, we used the ensemble mean of GPP estimates (for 2009-2010) generated using multivariate regression splines (MARS), random forests (RF), and artificial neural networks (ANN) available through the Data Portal of the Max Planck Institute for Biogeochemistry (https://www.bgc-jena.mpg.de). The model results are also compared with the multi-year average 3-hourly GPP and ecosystem respiration ($R_{eco}$) from the Boreal Ecosystem Productivity Simulator (BEPS) (Chen et al., 2012), which is normally used by GEOS-Chem in flux inversions (Deng et al., 2014, 2016). In BEPS, the annual terrestrial ecosystem exchange imposed in each grid box ($4° \times 5°$) is neutral (Deng and Chen, 2011) (i.e. GPP = $R_{eco}$). BEPS is driven by NCEP reanalysis dataset. Finally, the model results are evaluated using the a posteriori $CO_2$ fluxes from the CarbonTracker data assimilation system (Peters et al., 2007) (version CT2013B) available at http://carbontracker.noaa.gov. All datasets used in evaluating the model's results are re-gridded to the CLASS-CTEM grid.

We evaluate the results of the inversion analyses (described in section 3.3) using the GEOS-Chem model by comparing the a posteriori $CO_2$ fields to atmospheric $CO_2$ observations from the Total Carbon Column Observing Network (TCCON) from which the column-averaged dry-air mole fractions of $CO_2$ (XCO2) are retrieved (Wunch et al., 2011). TCCON data were obtained from the TCCON Data Archive, hosted by the Carbon Dioxide Information Analysis Center (CDIAC) (http://tccon.ornl.gov/). For the comparisons, we use observations from the current TCCON GGG2014 data set from 13 different sites (Table 2) (see also Deng et al. (2014)) in 2009 and 2010. We also evaluate the inversion analyses using aircraft data from the HIAPER Pole-to-Pole Observations (HIPPO) project (http://hippo.ornl.gov/). We use the 10-second averaged data from the HIPPO-1, HIPPO-2, HIPPO-3 campaigns (Wofsy, 2011; Wofsy et al., 2012), for 9 to 21 January 2009, 31 October to 22 November 2009, and 24 March to 16 April 2010, respectively.

## 3 Experimental Design

In this study, we performed a offline coupling between CLASS-CTEM and GEM-MACH-GHG. We first run GEM-MACH-GHG to produce the necessary meteorological variables required to drive CLASS-CTEM. Then, the simulated fluxes from CLASS-CTEM are used in GEM-MACH-GHG to simulate the $CO_2$ concentrations. Given the complexity of online coupling (which is harder to implement), the offline simulations provide an affordable means to better isolate and assess the sensitivity of the model to different climate forcings. Offline coupling is also a step toward online coupling.

When coupling CLASS-CTEM to GEM-MACH-GHG, we first identify a necessarily-imperfect spin-up procedure that transitions from climate data forcing from a standard dataset such as CRU-NCEP to a short sequence of operational meteorological analyses (Section 3.1). Once fluxes are available from CLASS-CTEM for CRU-NCEP meteorology with the standard spin-up procedure and from GEM-MACH-GHG with the modified spin-up procedure, the simulations of $CO_2$ that are performed with GEM-MACH-GHG are described in Section 3.2. Finally, the a priori fluxes from CLASS-CTEM are tested in a flux inversion experiment which is described in Section 3.3.

These different simulations will assess whether the deficiencies of CTEM-GEM prior fluxes would be evident in the context of flux inversion when observations can correct for prior flux errors, and how the deficiencies in CTEM-GEM prior fluxes compare to other prior fluxes. If they are consistent or no worse than other sources of prior fluxes, then they are a potential starting point for our flux estimation system.

### 3.1 CLASS-CTEM Runs

To test the sensitivity of the simulated carbon fluxes to the meteorological forcing, we performed a series of experiments with CLASS-CTEM using two different meteorological inputs from (1) CRU-NCEP (hereafter called CTEM-CRUNCEP), which has been used to drive CLASS-CTEM simulations in previous studies (Melton and Arora, 2014; Melton et al., 2015; Badawy et al., 2016), and (2) GEM-MACH-GHG (hereafter called CTEM-GEM). For the CTEM-CRUNCEP run, the model was first initialized (to represent the pre-industrial period 1861-1900) by running it to equilibrium using repeated 1901-1940 CRU-NCEP climate, a constant globally uniform $CO_2$ of 286.37 ppm, and a fixed vegetation fractional coverage corresponding to the year 1861 until carbon pools and fluxes were in steady state (zero mean annual net ecosystem exchange (NEE)). The model was then run from 1901-2010 using varying $CO_2$ concentrations and CRU-NCEP meteorology.

For the CTEM-GEM run, the meteorological inputs from GEM-MACH-GHG were only available for 2009-2010 at the time of this study, and hence no global climate data available for the pre-industrial run. In general, reanalysis output begins around 1949 (e.g. NCEP-NCAR reanalyses) when the observing system had sufficient coverage, and as noted earlier, analyses from operational systems are restricted to much shorter and recent periods because of the constant change in model, observations and assimilation schemes. Therefore, the spin-up simulation was performed with a constant uniform $CO_2$ concentration of 387.4 ppm (corresponding to 2009) and a fixed vegetation fractional coverage corresponding to the same year. The spin-up simulations were driven with repeated meteorological data for the 2009-2010 period until model pools reached equilibrium.

The transient simulation for the 2009-2010 period was then initialized from the spin-up simulations using varying $CO_2$ concentrations and GEM meteorology.

To assess the impact of using present climate to spin up the model on the simulated carbon pools and fluxes, we also performed a special run that used repeated meteorological data for 2009-2010 from CRU-NCEP, and constant uniform $CO_2$ of 387.4 ppm, and a fixed vegetation fractional coverage corresponding to the year 2009 until the model pools reach equilibrium (hereafter CTEM-CRUNCEP2yr).

Note that fire and land use change are not taken into account in the current model's simulations due to the large uncertainty in the global land use history (Houghton et al., 2012) that may yield significant biases in the simulated $CO_2$ fluxes. Also, the standard model parameters were not changed or tuned to improve model performance when using alternative meteorological inputs. Hence the main differences between the CLASS-CTEM runs are the meteorological inputs, and the set-up of the spin-up simulations.

## 3.2 Forward simulation using GEM-MACH-GHG model

Forward simulations are performed using the GEM-MACH-GHG model to evaluate how well CLASS-CTEM, using meteorological inputs from GEM-MACH-GHG, is able to reproduce temporal variations in atmospheric $CO_2$ at monitoring stations. The estimated NEE from CTEM-CRUNCEP and CTEM-GEM were used as a surface boundary condition in GEM-MACH-GHG, which transports the signal from the surface fluxes throughout the atmosphere, to validate the resulting modelled concentrations against observations of atmospheric $CO_2$. The other fluxes are kept the same as in Polavarapu et al. (2016). Specifically, the anthropogenic emissions from fossil fuel burning and cement manufacturing, biomass burning, ocean-atmosphere carbon exchange and initial atmospheric concentration (Jan 1, 2009) are based on CT2013B (Peters et al., 2007).

## 3.3 Inversion Analysis Configuration in the GEOS-Chem Model

Because flux inversions have been performed for over a decade with the in situ measurements, there is a considerable body of literature of such inversion results (e.g. Rödenbeck et al. (2003); Peters et al. (2007), and Peylin et al. (2013)). Consequently, for our experiments, we use this observing network as opposed to a combined one that includes the more recent satellite missions. Thus, the GEOS-Chem flux inversions use the flask observations of atmospheric $CO_2$ collected by NOAA ESRL Carbon Cycle Cooperative Global Air Sampling Network sites (Dlugokencky et al., 2015) and ECCC sampling sites (Worthy et al., 2009). We use the same set of observation sites as described in Deng et al. (2014) (see their Section 2.1.2).

In this study, we use the similar a priori $CO_2$ fluxes of the anthropogenic emissions from fossil fuel burning and cement manufacturing, biomass burning, and ocean-atmosphere carbon exchange described in Deng et al. (2014) in order to maximize comparability with the those results. However, for the biospheric flux of $CO_2$, we conducted three runs using three different NEE priors from CTEM-GEM, CTEM-CRUNCEP, and BEPS. The optimized 3-D $CO_2$ mixing ratio field from CarbonTracker was used as the initial $CO_2$ field in the inversion runs.This choice was made because the CarbonTracker initial state produced a smaller global mean bias than the GEOS-Chem initial state. This may indicate that the interhemispheric gradient in TM5 is better than that in GEOS-Chem. When assimilating the flask data, we use a threshold of $8 \, \mathrm{ppm}$ for rejecting observations in

the GEOS-Chem assimilations, and this means when the difference of observed and the modeled mixing ratio is larger than $8\,\mathrm{ppm}$, the observation is not ingested in the model and does not provide any information on the fluxes. We assimilated 5365, 5393, and 5601 observations into our model in inversions using CTEM-CRUNCEP, CTEM-GEM, and BEPS prior fluxes, respectively.

## 4 Results and Discussion

For the meteorological data, we compare temperature, shortwave radiation, and precipitation, which are considered to be the most important variables controlling land carbon dynamics (Piao et al., 2013). We also compare the component fluxes of GPP, $R_{\mathrm{eco}}$, and net ecosystem exchange (NEE = $R_{\mathrm{eco}}$ - GPP) in order to identify the potential drivers of differences between model simulations. To examine regional differences, data and model output are also spatially aggregated to the 11 land regions of the TransCom inverse model inter-comparison project (Gurney et al., 2003).

### 4.1 Differences in Meteorological Forcing

Here, we evaluate the meteorological data from GEM by comparing it against CRU-NCEP, CRU and ERAI datasets where possible. Figure 1 shows the spatial patterns of the differences in mean annual temperature (averaged over the period 2009-2010) between GEM, CRU-NCEP, and CRU. The differences between CRU-NCEP and CRU show cold biases in middle and high Northern latitudes and warm biases in Africa and South America. CRU-NCEP retains the monthly climatology of CRU but adds the daily and diurnal variations of NCEP reanalyses (Wei et al., 2014). Thus, differences in annual mean temperature of CRU and CRU-NCEP should be small by design. In contrast, GEM is warmer than CRU over the North high latitudes and generally cooler elsewhere. The comparison also shows that CRU-NCEP is cooler than GEM in Northeastern North America, Eastern Europe, and Eastern Asia, and warmer in Africa, Southwestern Asia, South America, and the west coastline of North America. The differences in Fig. 1c are much larger than those seen in Fig. 1a because GEM analyses are completely independent of CRU. NCEP reanalyses are constrained by the global meteorological observing system and the datasets used in 2009-2010 are likely broadly similar to that used by operational centers such as ECCC. Indeed, Zhao et al. (2006) compared meteorological fields from NCEP, the Data Assimilation Office (DAO) (currently called the GMAO), and ECMWF for the 2000-2003 period and found that the NCEP fields had a cold bias at all latitudes and that the bias was largest in the tropics, which is similar to the bias in the GEM fields. Zhao et al. (2006) also found that the ECMWF ERA-40 (the precursor to ERAI) and DAO fields had smaller zonal mean biases compared to NCEP, but the ERA-40 fields were similar to those from GEM in that they had a high bias at high latitudes.

To better illustrate the differences between the datasets, we have plotted in Fig. 2 the monthly mean temperature averaged for the 11 TransCom land regions. All the data show the same seasonal variations, with opposite phases of temperature between hemispheres. The largest differences are found in the tropics and the Southern hemisphere. GEM tends to be biased low compared to the other data in Northern Africa, Southern Africa and temperate South America. In these regions, ERAI is closer to CRU observations, but in tropical South America, GEM is closer to observations. Since ERAI is a reanalysis product, it is

expected to be much better than an operational analysis. Our GEM analyses are also further degraded (in terms of resolution) from ECCC operational analyses, so any reasonable comparability to ERAI results are considered promising. CRU-NCEP overall is in better agreement with the observations (CRU). This is not surprising given that CRU-NCEP was produced by combining CRU and NCEP/NCAR Reanalysis products.

Figure 3 shows the spatial distribution of the differences of mean annual shortwave radiation (averaged over the period 2009-2010) between GEM and CRU-NCEP, and ERAI datasets. Shortwave radiation estimates are not available in the CRU data set. The comparison indicates that CRU-NCEP is approximately 15-70 W m$^{-2}$ higher (sunny bias) than ERAI in the high latitudes and in the tropical land regions. In arid areas (i.e. Australia, Sahara, South Africa, Southern North America, Tibetan Plateau, and West Asia), CRU-NCEP is approximately 15-50 W m$^{-2}$ lower than ERAI. In contrast, GEM is approximately 10-

60 W m$^{-2}$ higher than ERAI over all land regions, with the highest values (40-60 W m$^{-2}$) over tropical lands. The shortwave radiation estimates from GEM is approximately 10-80 W m$^{-2}$ higher than those from CRU-NCEP over nearly all land regions, with the exception of Europe, Eastern North America and in a few grid cells in the tropical regions, where CRU-NCEP is higher (10-80 W m$^{-2}$).

Figure 4 shows the monthly mean shortwave radiation averaged for the TransCom land regions. GEM and ERAI have more

similar seasonal variability compared to CRU-NCEP in most of the land regions, especially in in the tropics. However, ERAI shows slightly lower monthly mean values in Eurasia regions, and in tropical South America. Zhao et al. (2006) also found that ERA-40 (the precursor to ERAI) underestimated shortwave radiation in the tropics. Differences between the datasets in the tropics may be due to cloudiness biases over the Intertropical Convergence Zone (ITCZ) region, which have a large impact on radiative forcing (Dee et al., 2011).

The comparisons of the differences in annual total precipitation between GEM, CRU-NCEP, and CRU datasets are shown in Fig. 5. The smallest differences are between CRU-NCEP and CRU. The largest differences in magnitude between CRU-NCEP and CRU are mainly in the tropics, particularly tropical Asia, and along the west coast of South America. Also, the largest differences between GEM and CRU are in the tropics. The comparison also indicates that CRU-NCEP is wetter than GEM in the tropics and sub-tropics, and in the temperate regions, but is drier than GEM in some areas of the boreal regions, and

over a few grid cells in central Africa, and China. In general, the tropics exhibit the largest differences between the GEM and CRU-NCEP datasets.

The comparisons between the monthly total precipitation integrated over the TransCom land regions are shown in Fig. 6. Unlike temperature and shortwave radiation (well represented by global models), there is a very clear difference in monthly total precipitation among the datasets, except between CRU-NCEP and CRU, which agree very well with some differences in

the tropics. It is clear that the largest differences occur mainly during summer in each hemisphere, which is associated with high precipitation. GEM tends to be drier mainly during summer. Despite the differences in the seasonal amplitude, GEM shows a quite similar seasonal variability compared to other datasets. We should keep in mind that precipitation estimates from the reanalysis/forecast systems are normally associated with large errors (Harris et al., 2014), particularly over land. These errors are due to problems with the convective parametrization in the models, and the fact that ground-based precipitation

observations are not yet used in the data assimilation systems. Also, CRU monthly precipitation suffers large biases in areas

where observations are sparse (i.e. tropics and southern Hemisphere) (Harris et al., 2014). In fact, the observation-based datasets are not based only on measurements, but are also sometimes model-dependent (i.e. filling gaps, interpolation, etc) (Harris et al., 2014). These deficiencies as well as the different spatial/temporal resolutions among models and observations can explain some of the differences between the datasets. Deficiencies in ERAI and CRU have been investigated in previous studies (Simmons
et al., 2010; Balsamo et al., 2010; Szczypta et al., 2011).

In summary, the meteorological fields from GEM are similar in quality to those from reanalyses (ERAI) and observation-based (CRU and CRU-NCEP) datasets. However, there are some notable discrepancies in seasonal variations and spatial distribution patterns between GEM and CRU-NCEP, particularly in precipitation estimates in the tropics, which will be reflected in the estimated carbon fluxes. Biases in precipitation may indicate that the convective scheme used in GEM system needs to
be improved, in particular, over the tropics. CLASS-CTEM driven by GEM precipitation will be impacted by these biases.

## 4.2  Impact of Meteorological Forcing on Carbon Fluxes

Here, we assess the impact of changing meteorological inputs on the simulated carbon fluxes to determine whether biases in fluxes can be attributed to biases in the meteorological variables.

### 4.2.1  Differences in Simulated Carbon Fluxes

To evaluate the spin-up procedure, the simulated global values of primary carbon pools and fluxes are summarized in Table 3 for the spin-up simulations from CTEM-CRUNCEP (which used the 1901-1940 climate data for the spin-up), and CTEM-GEM (which used the 2009-2010 climate data for the spin-up). CTEM-GEM produces smaller values of carbon pools and fluxes compared to CTEM-CRUNCEP. One possible explanation for this is the use of the present climate to spin up the model in the case of CTEM-GEM. Table 3 shows also the global values of carbon pools and fluxes simulated by the CTEM-CRUNCEP2yr
experiment, which also uses just the 2009-2010 climate to spin up the model. Rather than reducing the size of carbon pools, CTEM-CRUNCEP2yr produces much higher values compared to both CTEM-CRUNCEP and CTEM-GEM. Table 3 also compares the mean areal land precipitation globally as well as for the tropical land band (30°N-30°S) averaged over the 1901-1940 (for CTEM-CRUNCEP), and 2009-2010 (for CTEM-CRUNCEP2yr and CTEM-GEM). For the CRU-NCEP runs, 2009-2010 is wetter than the 1901-1940 period at global and tropical scales, which can explain the higher productivity in
CTEM-CRUNCEP2yr compared to CTEM-CRUNCEP. GEM precipitation (2009-2010) is slightly higher than CRU-NCEP (1901-1940) at the global scale, but lower over the tropical band for the same periods. The drier tropical band is reflected in the estimated tropical GPP from CTEM-GEM (Table 3), which dominates the global total GPP (Beer et al. (2010), Anav et al. (2015), and many others). This may explain the low carbon pools simulated by CTEM-GEM. This comparison suggests that precipitation plays a significant role in plant productivity in the tropics, and thus accurate precipitation patterns are necessary
to establish realistic initial values for carbon pools and fluxes during the spin-up runs. Despite the differences in model inputs and spin-up configuration, the initial global carbon pools and fluxes from CTEM-GEM, however, are still within the range of the preindustrial values from other modelling studies listed in Melton and Arora (2014, Table 2).

The low GPP values from CTEM-GEM warrant further discussion given that the initial estimates of carbon pools and fluxes are critical to obtain an accurate estimate of historical $CO_2$ fluxes (Exbrayat et al., 2014; Tian et al., 2015). Carbon stocks are often not well modelled in TEMs (Houghton et al., 2012; Tian et al., 2015). The modelled pool sizes can be adjusted by tuning the model parameters in order to match observation-based estimates of carbon stocks. For example, Carvalhais et al. (2008,
2010) have reported the limitation of the carbon cycle steady state assumption in TEMs. Carvalhais et al. (2010), therefore, introduced a new parameter in the CASA model that forced the adjustment of both vegetation and soil carbon pools from equilibrium (after spin-up) allowing for model runs to be initialized either as net sinks or sources. They found that including this new parameter yielded better model performance in simulating carbon fluxes in comparison to observations. Moreover, their modelled soil carbon stocks became closer to observations. However, large uncertainties and errors in measurements can
produce biased parameters, and hence poor model performance. Thus, forcing agreement to a given global mean value of GPP (e.g. 120 PgC year$^{-1}$) by tuning model parameters may lead to worse model performance and is not justifiable given the observational uncertainty in this value. Given that CLASS-CTEM will provide only a prior estimate of NEE (not GPP and $R_{eco}$ separately, at least in the first stage) for flux inversions in EC-CAS, adjusting the initial carbon pools modelled by CLASS-CTEM is not necessary and would not likely change the major conclusions derived here. Moreover, tuning of CLASS-CTEM
specifically for the far-from-ideal spin-up process that we employed for the GEM fields would be dubious and would make comparison with CTEM-CRUNCEP results difficult. Beyond the global budget, which is well constrained by atmospheric data (Peylin et al., 2013), the main focus is to assess the ability of the model to simulate the spatial and temporal flux variations (mainly the NEE that is used in flux inversions) in response to changes in environmental conditions and its ability to match the atmospheric signal.

For transient simulations, the simulated terrestrial carbon fluxes from the two simulations (CTEM-CRUNCEP and CTEM-GEM) are compared to each other as well as to observation-based estimates (where possible) or independent model results. Figure 7 shows the annual spatial difference of GPP simulated by CLASS-CTEM (averaged over the period 2009-2010) and the observation-based GPP estimates from FLUXCOM. The figure also shows the spatial difference between the modelled GPP from BEPS and FLUXCOM and the zonal distribution of GPP from all datasets. There are significant differences in the
annual GPP between the two simulations (CTEM-CRUNCEP and CTEM-GEM) and the evaluation data, particularly in the most highly vegetated areas (i.e. the tropics, and the boreal and temperate regions). CTEM-CRUNCEP and CTEM-GEM have similar spatial differences over Western Europe and boreal Asia, and to a lesser extent over North America but they show poor agreement in the tropics. Tropical GPP from CTEM-CRUNCEP is overestimated compared to FLUXCOM (see also Fig. 7b). In contrast, it is underestimated in the Amazonian region, western Africa, and tropical Asia with CTEM-GEM. In comparison
to other model results, the spatial distribution of the difference between BEPS and FLUXCOM (Fig. 7a) reveals relatively smaller different patterns compared to CTEM. The zonally averaged GPP in Fig. 7b indicates that CTEM-CRUNCEP and BEPS agrees very well with FLUXCOM compared to CTEM-GEM, which underestimates GPP in the tropics.

Since the formulation of most models, including CLASS-CTEM and BEPS, links respiration to photosynthesis (Melton and Arora, 2016), $R_{eco}$ estimates from both simulations and BEPS show a similar pattern to GPP (spatially and zonally), with
significant differences in the most productive ecosystems (not shown here). The large discrepancies in seasonal variations and

spatial distribution patterns between GEM and CRU-NCEP are due to the precipitation differences (temperature and shortwave radiation have much better agreement), particularly in the tropics. Figures 5 and 7 suggest that the differences in the spatial pattern of GPP are more closely associated with precipitation than temperature or shortwave radiation differences over the tropics. This is consistent with previous findings (Nemani et al., 2003; Jung et al., 2007; Beer et al., 2010; Piao et al., 2013; Anav et al., 2015) that interannual variation of productivity is primarily correlated with the precipitation over the tropics.

To examine regional differences, the seasonal variation and the annual mean of GPP from both simulations, and BEPS are also spatially aggregated to the 11 TransCom land regions and compared to FLUXCOM in Fig. 8. For $R_{\text{eco}}$ (not shown here), the same conclusions can be drawn as from the GPP figures. Figure 8 shows that the seasonal variations of GPP from CTEM-CRUNCEP are consistently higher than those from CTEM-GEM. In the northern hemisphere regions, flux estimates have large seasonal variations, i.e., small values in winter and high values in summer, reflecting the seasonal change in carbon uptake by the land vegetation. The largest differences between CTEM-CRUNCEP and CTEM-GEM, in terms of the amplitude of the seasonal cycle, are found in the tropics, with CTEM-GEM having smaller amplitudes. However, in tropical Asia, the seasonal cycle from CTEM-GEM agrees well with BEPS and FLUXCOM compared to CTEM-CRUNCEP, which has larger annual GPP and $R_{\text{eco}}$ (not shown here). Also, CTEM-GEM agrees very well with FLUXCOM in Europe and North America boreal compared to CTEM-CRUNCEP, especially during the growing season. In general, CTEM-GEM have some differences compared to CTEM-CRUNCEP over all regions mainly in terms of the amplitude, and to a lesser extent in the phase of the seasonal cycle. This is consistent with the findings of Dalmonech et al. (2015) who tested the impact of coupled and uncoupled configurations of JSBACH land surface component of the Max Planck Institute Earth System Model (MPI-ESM) on the simulated land carbon fluxes. They found that biases in the meteorological forcing to a large extent control the magnitude of GPP rather than the phenology and seasonal cycle of productivity, which could be more related to the model formulations (i.e. the timing and length of the growing season). In summary, these results indicate that the simulated fluxes from CLASS-CTEM are sensitive to the meteorological forcings over all land regions, which is consistent with the finding of Garnaud et al. (2014).

The annual GPP, $R_{\text{eco}}$, and the net flux are given in Table 4. Annual GPP values from CTEM-GEM for 2009 and 2010 are smaller than the multi-year average of GPP from BEPS (119.5 PgC) (Deng et al., 2014), and from Beer et al. (2010) (123 $\pm 8$ PgC). Annual GPP values from CTEM-CRUNCEP for 2009 and 2010 are higher than those from BEPS and the calculated GPP from FLUXCOM (119.83 and 120 PgC for 2009, 2010 respectively). This leads to a stronger land carbon sink from CTEM-CRUNNCEP compared to CTEM-GEM (Table 4). The weaker sink in CTEM-GEM is due to the lower precipitation estimates in the tropics (the region that mainly controls interannual variability in the carbon cycle), and hence lower global GPP (Piao et al., 2013; Beer et al., 2010). Previous studies have reported a wide range of the global terrestrial GPP. For example, Piao et al. (2013) showed that GPP averaged across 10 land models is 133 $\pm 15$ (ranging from 111 $\pm 4$ PgC to 151 $\pm 4$ PgC). Other studies that are based on carbon cycle data assimilation suggest a GPP around 150 PgC (Koffi et al., 2012; Peylin et al., 2016), similar to Welp et al. (2011).

To assess the impact of the differences in GPP and $R_{\text{eco}}$ from CTEM-CRUNCEP and CTEM-GEM on the seasonal cycle of NEE (the difference between GPP and $R_{\text{eco}}$), Fig. 9 compares the NEE seasonal cycle from both simulations with the

simulated prior NEE from BEPS (multi-year average) and the optimized NEE from CT2013B for 2009 and 2010 over the TransCom land regions. BEPS produces the smallest amplitude of the seasonal cycle of NEE while CTEM-CRUNCEP has the largest amplitude in northern land regions, except boreal Eurasia where the optimized NEE from CT2013B exhibits the largest amplitude (Fig. 9). For the South American tropical region, all models show considerable disagreement in the seasonal cycle, sometimes with opposite phases. In the northern hemisphere, CTEM-GEM and CTEM-CRUNCEP have better agreement with each other during winter than in summer. CTEM-GEM also tends to have the peak of the growing season one month earlier than CTEM-CRUNCEP (i.e. Eurasian boreal and North America temperate) due to the differences in GPP seasonal cycle (see Fig. 8). Even though there is large difference in the amplitude of the seasonal cycle of GPP (the same for $R_{\text{eco}}$ - not shown here) from CTEM-GEM compared to CTEM-CRUNCEP (Fig. 8), the difference is much smaller in NEE. This is due to the fact that NEE is the difference between two large terms (GPP and $R_{\text{eco}}$). That means, even though GPP and $R_{\text{eco}}$ have large biases compared to observation-based estimates over the tropics, the biases in NEE are much smaller (4).

A recent study by Byrne et al. (JGR, under review, personal communication) evaluated the fluxes from CTEM-GEM and CTEM-CRUNCEP (the same fluxes used in the current study) in northern mid-latitude ecosystems by comparing GPP against Global Ozone Monitoring Experiment-2 (GOME-2) solar induced fluorescence (SIF) (Joiner et al., 2013; Khler et al., 2015) and NEE against total column $CO_2$ ($XCO_2$) from TCCON (Wunch et al., 2011). They found that while GPP from CTEM-CRUNCEP showed closer agreement with SIF than CTEM-GEM, CTEM-GEM showed closer agreement with TCCON $XCO_2$ than CTEM-CRUNCEP. This implies that the biases in GPP might be compensated by biases in $R_{\text{eco}}$, which result in improved NEE fluxes from CTEM-GEM.

The NEE values in Table 4 show that CTEM-CRUNCEP tends to have a higher carbon sink compared to CTEM-GEM. CTEM-GEM simulates a land carbon sink of -1.2 and -2.2 PgC year$^{-1}$ for 2009 and 2010, respectively, which is close to the optimized land sink from the CCDAS study of Peylin et al. (2016) (around 2.2 PgC year$^{-1}$, for the 2000-2009 period). NEE from CTEM-GEM is also more compatible with the Global Carbon Budget (GCP) (Quéré et al., 2015) (2.4 $\pm$0.8 PgC year$^{-1}$ for the 2000-2009 period).

In summary, CTEM-GEM has some issue in the tropics but that is where all prior (and posterior flux estimates) disagree and where observations are also inconsistent. However, Figs. 8 and 9 show that the Northern Hemisphere (where CTEM-GEM is fairly consistent with the other estimates) dominate the global seasonal cycle of carbon fluxes because it has the largest land areas that mainly dominated by forest ecosystems. At the same time, the net contribution from the tropical and the Sothern Hemisphere regions are close to zero due to their opposite (and relatively small) seasonal cycles. Accordingly, the prior NEE information from CTEM-GEM is considered to be suitable for testing in the data assimilation context. However, flux estimates in the tropics from CTEM-GEM should be treated with caution.

### 4.2.2 Modelled $CO_2$ Concentration

To assess the quality of the $CO_2$ fluxes from CLASS-CTEM simulations, terrestrial NEE fluxes from CTEM-CRUNCEP and CTEM-GEM are used as a priori land fluxes in the GEM-MACH-GHG global atmospheric $CO_2$ transport model. For comparisons, GEM-MACH-GHG was also run using the posterior NEE fluxes from CT2013B as described in Polavarapu et al.

(2016). In these forward simulations, the anthropogenic emissions from fossil fuel burning and cement manufacturing, biomass burning, ocean-atmosphere carbon exchange are based on CT2013B (Peters et al., 2007) so that the only difference between the three runs is the terrestrial NEE fluxes.

Figure 10 compares the monthly times series of the modelled $CO_2$ from the two CLASS-CTEM simulations with that based on the CT2013B posterior fluxes, and observed $CO_2$ at selected sites that are representative of various global regions (listed in Table 1) for continuous $CO_2$ measurements (Worthy et al., 2009; Dlugokencky et al., 2015). At all observation sites, the simulations forced with CTEM-CRUNCEP and CTEM-GEM NEE fluxes (green and blue curves) have a similar overestimation of the observed atmospheric $CO_2$ from December to May, but the simulation forced with CT2013B fluxes (red curves) has a much better match to observations. This makes sense because CT2013B fluxes have been informed by atmospheric observations whereas the other two fluxes have not. Nevertheless, the overestimation of the $CO_2$ concentrations can be attributed to the smaller net uptake simulated by CLASS-CTEM in the Northern Hemisphere (NH) regions during the winter season (see Fig. 9). At ALT and BRW, the two CLASS-CTEM simulations, in particularly CTEM-CRUNCEP, show a 1 month shift of the peak of the growing season compared to the observations. These differences in the seasonal cycle might indicate some limitations in the phenology of CLASS-CTEM (i.e. the larger source of carbon fluxes in wintertime and the seasonal cycle shift) and should be further investigated in future studies. Interestingly, at ALT, BRW, IZO, MLO, and ZEP, during the NH growing season, in particularly in 2009, CTEM-GEM has a better match to the observation than CT2013B and CTEM-CRUNCEP. This is promising as it indicates that the CTEM-GEM fluxes may be suitable to be used in the global inversion system (EC-CAS).

The large differences in the modelled $CO_2$ concentrations from the two CLASS-CTEM simulations highlight the sensitivity of the transport model to land fluxes. This might contradict the finding of (Ott et al., 2015) that large differences in prior flux estimates result in only small differences in $CO_2$ concentrations. The autumn underestimation with CT2013B fluxes (e.g. at ALT, BRW, and IZO) is likely due to a mismatch in seasonal-scale meridional transport between GEM-MACH-GHG and TM5 (the model used to produce CT2013B) (Polavarapu et al., 2016). We should also keep in mind that the performance of the forward simulations also depends on the site specific conditions (e.g. topographic features, local sources of emissions, its location from urban areas, accuracy of the measurements, etc.) (Peters et al., 2005; Niwa et al., 2012; Shirai et al., 2017) and also on the performance of the transport model (Gurney et al., 2003, 2004).

To better assess the quality of the modelled $CO_2$ concentrations obtained with the two CLASS-CTEM simulations, Fig. 11 shows the Taylor diagram (Taylor, 2001) that compares the modelled $CO_2$ from CTEM-CRUNCEP, CTEM-GEM, and CT2013B with the observed $CO_2$ at the selected sites (listed in Table 1). The position of each dot appearing on the plot quantifies how closely the modelled concentrations match the observations. The centered root mean square (RMS) difference between the simulated and observed patterns is proportional to the distance to the point on the x-axis identified as "1.0", which is the observations. The normalized standard deviation (dividing the standard deviation of simulated by the standard deviation of the observed) of the simulated pattern is proportional to the radial distance from the origin and it represents the agreement in the amplitude of the variability between the modelled and observed concentrations. As mentioned before, the simulation forced with the optimized fluxes from CT2013B has a much better match to observations. Also, Fig. 11 shows that CTEM-

GEM agrees better with the observations at all selected sites compared to CTEM-CRUNCEP in terms of correlations and the amplitude of the variability, which indicates an overall reasonable performance of the CTEM-GEM, especially during the growing season as mentioned before (see Fig. 10).

## 4.3 Inversion Analyses

The results in the previous two sections revealed that the GEM-MACH-GHG simulation of atmospheric $CO_2$ using CTEM-GEM fluxes is able to reproduce temporal variations in atmospheric $CO_2$ at the selected sites. Since CTEM-GEM will be used as the land component of EC-CAS, which is presently under development and thus not yet available, here we use the GEOS-Chem data assimilation system to examine the impact on regional flux estimates of using CTEM-GEM and CTEM-CRUNCEP as prior fluxes in the context of an atmospheric $CO_2$ inversion analysis. To determine how the retrieved fluxes obtained with

the two CTEM-based priors compare to other documented inverse modelling results, we also perform an inversion analysis using BEPS prior fluxes, which is the ecosystem model used in the GEOS-Chem inversions of Deng et al. (2014, 2016), and we compare our results to the retrieved fluxes from CT2013.

### 4.3.1 Seasonal Cycle of the Flux Estimates

Figure 12 shows the seasonal cycle of the a posteriori NEE from the GEOS-Chem inversion analyses using the three different a
priori estimates of NEE (CTEM-GEM, CTEM-CRUNCEP, and BEPS), together with the optimized NEE from CT2013B. The a priori NEE from CTEM-GEM, CTEM-CRUNCEP and BEPS are also shown. For northern land, there are some differences between the optimized NEE in terms of the amplitude and the growing season, especially in temperate region and in boreal Eurasia where the peak carbon uptake is greatest for CT2013B. However, the spread in the posterior estimates for most northern land regions is smaller (compared to the tropical regions) because the surface observation network can reasonably constrain the

northern extratropical latitudes (Peylin et al., 2013). The spread between the fluxes is larger in the tropical regions, where NEE seasonal cycles show less agreement in both phase and magnitude. There are large differences in the seasonal cycle between the CTEM-based fluxes (CTEM-CRUNCEP and CTEM-GEM) and the BEPS-based fluxes in northern Africa and tropical South America that are present in both the prior and posterior fluxes. As a result of the limited observational coverage in the tropics, the posterior fluxes are strongly influenced by the prior fluxes. Consequently, the differences in the prior fluxes across

the inversions are reflected in the posterior fluxes. Similarly, in the southern extratropics the posterior fluxes primarily reflect the prior flux distributions due to the sparsity of observations. This result is consistent with that of Peylin et al. (2013) who also found more disagreement of various inversion results in the tropics and southern hemisphere.

The amplitude of seasonal cycle of the optimized NEE from CTEM-CRUNCEP is significantly reduced compared to the a priori seasonal cycle in almost all land regions. The changes in the amplitude of the seasonal cycle for CTEM-GEM are smaller

than those for CTEM-CRUNCEP. Figure 9 shows that CTEM-CRUNCEP tends to have a larger amplitude of NEE compared to CTEM-GEM and the evaluation data in all regions. Also, the comparison in Figs. 10 and 11 revealed that modelled $CO_2$ concentrations from CTEM-GEM has a much better match to observations compared to CTEM-CRUNCEP, especially during the growing season. This might explain the significant shift in the optimized NEE from CTEM-CRUNCEP compared to CTEM-

GEM. This also might indicate that the simulated fluxes from CTEM-GEM are more consistent with the atmospheric $CO_2$ signal than CTEM-CRUNCEP. As noted above, the recent study by Byrne et al. (JGR, under review, personal communication) evaluated the a priori fluxes from CTEM-GEM and CTEM-CRUNCEP and found that compared to TCCON data, CTEM-GEM NEE provided a better simulation of the atmospheric $CO_2$ seasonal cycle. They found that CTEM-CRUNCEP NEE produced a seasonal cycle in atmospheric $CO_2$ with a larger amplitude and with the onset of the springtime drawdown delayed by 10 days relative to CTEM-GEM NEE. However, both CLASS-CTEM simulations tend to have a large carbon source in wintertime, which is subsequently reduced in the optimized NEE (Fig. 12), especially in Europe, and NH temperate regions. This demonstrates the capacity of inversion systems to constrain the phenological cycle in CLASS-CTEM, and ultimately can be used to optimize model parameters.

### 4.3.2   Annual Mean Flux Estimates

The total annual a priori and a posteriori NEE from the GEOS-Chem inversion analyses for 2009-2010 are shown in Fig. 13 for the 11 land TransCom regions, along with the optimized values from CT2013B. Note that the tropical Asia panel has a different scale. All models estimate a sink (both for the a priori and the a posteriori) for the North America temperate, South American tropical, and Eurasian regions (except for temperate Eurasia, which has a source for CTEM-GEM prior). The largest difference between the a priori and a posteriori NEE in terms of the sign and magnitude were obtained for the Southern American temperate and Northern and Southern African regions. This is due to the fact that the tropics and Southern Hemisphere are poorly constrained by the current $CO_2$ network. Although the CT2013B fluxes tend to have stronger uptake in boreal Eurasia and the two African regions, BEPS has the largest uptake in the extra-tropical regions (North America temperate, Europe, Eurasia temperate) and tropical Asia. The two CLASS-CTEM simulations show large differences, from each other, in the optimized NEE in all land regions. Some of the difference between CT2013B and all of the GEOS-Chem estimates can be attributed to different transport models and configurations of data assimilation (Peylin et al., 2013) used in GEOS-Chem and CT2013B (uses Tracer Transport Model - version 5 (TM5) (Peters et al., 2007)). Figure 13 also indicates that, for 2009, GEOS-Chem allocates the strongest sink in Tropical Asia for CTEM-CRUNCEP (2 PgC year$^{-1}$). Given that the inversion with CTEM-CRUNCEP fluxes suggest much weaker sinks (or larger sources) for the South American Tropical and Northern African regions compared to CTEM-GEM and BEPS, we suspect that stronger uptake in tropical Asia could reflect the inversion compensating for the larger sources in tropical south America and northern Africa. The hypothesis is that CTEM-CRUNCEP start with a larger a priori sink in tropical Asia and it gets enhanced in the inversion to compensate for the other tropical regional biases.

Figure 14 shows the global annual totals of NEE (a priori and a posteriori) for 2009-2010, as well as annual totals, aggregated into three latitudinal bands: Southern Hemisphere (SH):90 S - 30° S, Tropics (TR): 30° S - 30° N, and Northern Hemisphere (NH): 30° N - 90° N. At the global scale, there is a good agreement between the optimized NEE in terms of magnitude. This indicates that the observations sufficiently constrain the global carbon budget so that the choice of prior fluxes is not critical. This is in agreement with Bruhwiler et al. (2011) who examine the impact of changing observation networks on flux estimates. However, optimized NEE for the three latitudinal bands show some differences, mainly in the tropics, where the

observational coverage is poor. These results are consistent with previous findings (e.g., Peters et al., 2007; Miller et al., 2015), which showed that optimized $CO_2$ fluxes in inversion analyses are heavily influenced by the spatial patterns in the a priori $CO_2$ fluxes, particularly in regions where observations are sparse (i.e. tropics and southern Hemisphere).

### 4.3.3    Evaluation of the Inversions

As described in section 3.3, we assimilated 5365, 5393, and 5601 observations into the inversion analysis using CTEM-CRUNCEP, CTEM-GEM, and BEPS prior fluxes. Based on these varied numbers of assimilated observations, we obtained $\chi^2$ (chi-squared, which is a measure of the consistency of actual model-data mismatch with prescribed error covariances of model-data mismatch and where values close to 1.0 indicate consistency) for the three inversions are 0.91, 0.89, 0.76, and the correlation coefficients between observations and modeled root mean square (RMS) are 0.8398, 0.8436, and 0.8529, respectively, for CRUNCEP, CTEM-GEM, and BEPS prior fluxes, respectively. The $\chi^2$ and the correlation coefficient values suggests that the performance of CTEM-GEM and CTEM-CRUNCEP are quite similar and reliable in terms of fitting the assimilated observations. Figure 15 shows the histogram of the residuals between modelled (using CTEM-CRUNCEP, CTEM-GEM, and BEPS prior fluxes) and observed $CO_2$ concentrations. It is clear that there is no significant differences in the shape or width of the frequency distributions. Also, the standard deviations ($\sigma$) are similar and the biases ($\mu$) are all small relative to the standard deviations biases for all simulations, indicating that all 3 experiments fit the assimilated data about equally well.

The flask observations from surface stations, which were assimilated by GEOS-Chem in the inversion, only provide a check on the consistency or set-up of the assimilation system. We still need to compare to independent observations that were not assimilated for validation. To more effectively evaluate the assimilation results, we compare the a posteriori $CO_2$ fields with independent data that were not ingested in the assimilation. Listed in Table 5 are the mean differences and root-mean-square errors (RMSEs) of the a posteriori $CO_2$ relative to TCCON data between July 2009 and June 2010. All three fluxes reproduce the TCCON data well, but the BEPS-based $CO_2$ fields have the smallest mean difference, RMSE, and inter-station bias of 0.49 ppm, 1.23 ppm, and 0,49 ppm, respectively. We find that the RMSE for the $CO_2$ fields based on the CTEM-GEM and CTEM-CRUNCEP fluxes are identical. We also find that the fields based on CTEM-GEM have a smaller inter-station bias of 0.53 ppm, whereas those based on CTEM-CRUNCEP have a smaller mean difference of 0.53 ppm.

We also compare the a posteriori $CO_2$ fields with HIPPO aircraft data (see Table 6). As with the comparison to TCCON data, we find that the a posteriori fields based on BEPS fluxes produce the smallest mean difference and RMSE relative to all of the aircraft data (between 0 - 10 km and 60°S - 60°N), with the fields from CTEM-GEM fluxes producing smaller RMSEs than those obtained from the CTEM-CRUNCEP fluxes. However, the differences in the RMSE between CTEM-CRUNCEP and CTEM-GEM are not statistically significant. Overall, the comparisons of the a posteriori results to the independent data indicate that CTEM-GEM can provide useful a priori fluxes for $CO_2$ inversion analyses.

## 5 Conclusions

CLASS-CTEM will be used to provide first-guess (a priori) terrestrial fluxes for the Environment Canada Carbon Assimilation System (EC-CAS) (Polavarapu et al., 2016). The transport model of EC-CAS that relates surface fluxes to atmospheric $CO_2$ concentrations is based on the GEM-MACH-GHG model (Polavarapu et al., 2016). To ensure consistency between the land and transport model, CLASS-CTEM will be driven by the standard meteorological forcing simulated (24 h forecast) by GEM-MACH-GHG. Therefore, the main focus of this study was to assess the impact of using the meteorological inputs from GEM-MACH-GHG in simulating both regional and global carbon fluxes by CLASS-CTEM.

We first evaluated the quality of the meteorological inputs from GEM-MACH-GHG against the standard meteorological forcing (CRU-NCEP) that is used to drive the latest versions of CLASS-CTEM, as well as against observation-based or reanalysis datasets. The comparison shows that radiation and temperature data from GEM-MACH-GHG and CRU-NCEP are in good agreement. However, there are some notable discrepancies between GEM and CRU-NCEP in terms of seasonal variations and spatial patterns of precipitation estimates, especially in the tropics, with GEM being drier than CRU-NCEP, ERA-Interim, and CRU. That might indicate that the convective scheme used in GEM-MACH-GHG system needs to be improved in particular over the tropics.

Fluxes produced with GEM meteorology were obtained using a modified spin-up procedure based on current climate only. While it is clearly unsatisfactory to use a short climatology to spin-up carbon pools, it is an inevitable problem when coupling a TEM to an assimilation system since the latter focus on only a few years at a time. Moreover, operational weather assimilation systems are constantly changing so long datasets of analyses are simply not possible to obtain, unless reanalysis datasets are used. The differences in the precipitation fields between GEM-MACH-GHG and CRU-NCEP was reflected in the estimated carbon fluxes (GPP and $R_{eco}$). The amplitude and, to a lesser extent, the phase of the seasonal cycle are different between the two simulations, especially in the tropics. This is consistent with the findings of Dalmonech et al. (2015), who found that meteorological biases significantly control the magnitude of the productivity rather than the phenology and the seasonal cycle of carbon fluxes. Overall, the differences in the simulated fluxes between the two CLASS-CTEM simulations indicate that the model is sensitive to the meteorological forcings over all land regions, and agree with the finding of Garnaud et al. (2014). Despite the deficiencies in the spin-up procedure, CTEM-GEM simulates a land carbon sink of -1.2 and -2.2 PgC year$^{-1}$ for 2009 and 2010, respectively, which is close to the optimized land sink from the CCDAS study of Peylin et al. (2016) (around 2.2 PgC year$^{-1}$, for the 2000-2009 period), and also compatible with the Global Carbon Budget (GCP) (Quéré et al., 2015) (2.4 $\pm$0.8 PgC year$^{-1}$ for the 20002009 period). However, flux estimates over the tropics from CTEM-GEM should be treated with caution due to the negative biases in the precipitation fields compared to all other datasets (i.e. ERA-Interim, CRU, and CRU-NCEP).

To assess their ability to model $CO_2$ at monitoring stations, NEE fluxes from CTEM-CRUNCEP and CTEM-GEM were used as a priori land fluxes in the GEM-MACH-GHG global atmospheric $CO_2$ transport model. The comparison shows that the modelled $CO_2$ concentrations forced with CTEM-GEM have a better match to the observations of $CO_2$ concentration during the NH growing season compared to CTEM-CRUNCEP. The differences in the modelled $CO_2$ concentrations from the two

CLASS-CTEM simulations highlight the sensitivity of the transport model to land fluxes. The results also provided insights into the deficiencies in CLASS-CTEM. For example, the comparison indicated that CTEM-CRUNCEP and CTEM-GEM NEE are overestimated compared to observations at all the selected sites during the NH winter season. The overestimation of the $CO_2$ concentrations can be attributed to the larger carbon source simulated by CLASS-CTEM in the Northern Hemisphere (NH) regions in wintertime. The results also show that the two CLASS-CTEM simulations have some difficulties in capturing the phase of seasonal cycle of the observations, which indicates deficiencies in the phenology scheme of CLASS-CTEM and this should be further investigated in future studies. The deficiencies in simulating the seasonal cycle was also noticed in the study by Arora et al. (2009), who compared simulated monthly $CO_2$ from CanESM1 (CTEM used as the land component of that model) against observations at selected sites and found that there was a shift in the seasonal cycle (about a month later) at Barrow, Niwot Ridge, and Mauna Loa (see their Figure 11). The study by Anav et al. (2013), which compared 18 Earth system models, also showed that CanESM2 has some limitations reproducing the net uptake of carbon during spring and summer months.

To examine the impact of using fluxes from CTEM-GEM and CTEM-CRUNCEP as a priori flux estimates in atmospheric inversion analyses, we used the GEOS-Chem data assimilation system since EC-CAS is still under development. We assimilated in situ atmospheric $CO_2$ observations from the surface network to estimate optimized monthly mean NEE fluxes for 2009-2010. The time series of the estimated fluxes, integrated over different land regions, revealed that the optimized NEE is shifted from its a priori pattern in order to fit the data. For comparison with the CTEM-based fluxes we also used BEPS fluxes (Deng et al., 2014) as an a priori in the inversion analyses. We found that the CTEM-based optimized fluxes produced atmospheric $CO_2$ concentrations that were consistent with those based on BEPS and they fit the assimilated data about equally well. The results are promising for the EC-CAS project as they demonstrate that the CTEM-GEM fluxes can provide useful a priori fluxes for the global inversion system.

By coupling CLASS-CTEM into EC-CAS, this study helps to pave the way for the coupled meteorological and ecosystem model within the EnKF (e.g. see conclusions of Miller et al. (2015)) and is considered an important step toward understanding how meteorological uncertainties affect both $CO_2$ flux estimates and modelled atmospheric transport. Ultimately, such an approach will provide more direct feedback to the CLASS-CTEM developers, and thus help to improve the performance of CLASS-CTEM by identifying the model limitations based on atmospheric constraints. This can also lead to improvements in the CanESM which is used to address the question of the feedback between climate change and the carbon cycle.

*Code and data availability.* Fortran code for CLASS-CTEM modelling frame-work is available on request and upon agreeing to ECCC's licensing agreement available at: http://collaboration.cmc.ec.gc.ca/science/rpn.comm. Please contact the coauthor, Joe Melton (joe.melton@ canada.ca), to obtain model code. The GEM and GEM-MACH source codes are integrated into the unique operational computing environments of ECCC. These source codes are copyrighted but are available upon request subject to the GNU Lesser General Public License (LGPL v2.1) agreement (contact the coauthor, Michael Neish (Michael.Neish@canada.ca)). Some documentation on GEM is available at: http://collaboration.cmc.ec.gc.ca/science/rpn/gem/gemdm/gemdm.html and http://collaboration.cmc.ec.gc.ca/science/rpn/gef_html_public/.

ECCCs model output data are available at: https://weather.gc.ca/grib/index_e.html. The GEOS-Chem model, including the adjoint code, is freely available to the public and is distributed through GitLab. Instructions for obtaining and running the model are available on the GEOS-Chem wiki: (http://wiki.seas.harvard.edu/geos-chem/). The NOAA in situ CO2 observations used in the inversion analysis are available from ftp://aftp.cmdl.noaa.gov/data/trace_gases/co2/flask/surface/. All data generated by CLASS-CTEM is available from ECCC upon completion of a licensing agreement.

*Author contributions.* Bakr Badawy and Saroja Polavarapu designed the experiments. Bakr Badawy generated the required meteorological fields for CLASS-CTEM from GEM-MACH-GHG and performed the simulations for CLASS-CTEM and GEM-MACH-GHG and analysed the experiments and created the figures. Feng Deng performed the simulations for GEOS-Chem in coordination with Dylan Jones who contributed to the analysis and the interpretation of the assimilation results. Michael Neish developed and contributed to the GEM-MACH-GHG model diagnostics. Joe Melton and Vivek Arora provided the code of CLASS-CTEM used in this study and provided input on the implementation of the model. Ray Nassar provided helpful discussions. Bakr Badawy prepared the manuscript with contributions from all co-authors.

*Competing interests.* The authors declare that they have no conflict of interest.

*Acknowledgements.* Bakr Badawy was supported by Environment and Climate Change Canada (ECCC). We gratefully acknowledge all the data providers including the NOAA CarbonTracker assimilation system team for making their model products publicly available at http://carbontracker.noaa.gov. The NOAA in situ $CO_2$ observations used in the inversion analysis are available from https://www.esrl.noaa.gov/gmd/dv/data/. We thank all the data providers who made their data available at the Global Atmosphere Watch (GAW) observing network. We also would like to thank Doug Worthy for developing and maintaining ECCCs greenhouse gas measurement network and for providing the $CO_2$ concentration measurements. We gratefully acknowledge TCCON PIs (cited in Table 2) for making their data available. TCCON data were obtained from the TCCON Data Archive, hosted by the Carbon Dioxide Information Analysis Center (CDIAC) - tccon.onrl.gov. We gratefully thank the HIPPO (HIAPER Pole-to-Pole Observations, National ScienceFoundation, NSF, NSF/NCAR Gulfstream-V (GV)) science team for making their data freely available at http://hippo.ornl.gov. The collection of the original HIPPO data were supported by NSF and NOAA. ERA-Interim data used in this study have been obtained from the ECMWF data server: http://data.ecmwf.int/data. We thank M. Jung for providing access to the GPP FLUXCOM data. We thank Douglas Chan who evaluated the manuscript for the internal review process at ECCC and provided helpful comments, which improved the manuscript. We would like to thank the two anonymous referees for insightful comments that helped to improve the paper.

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

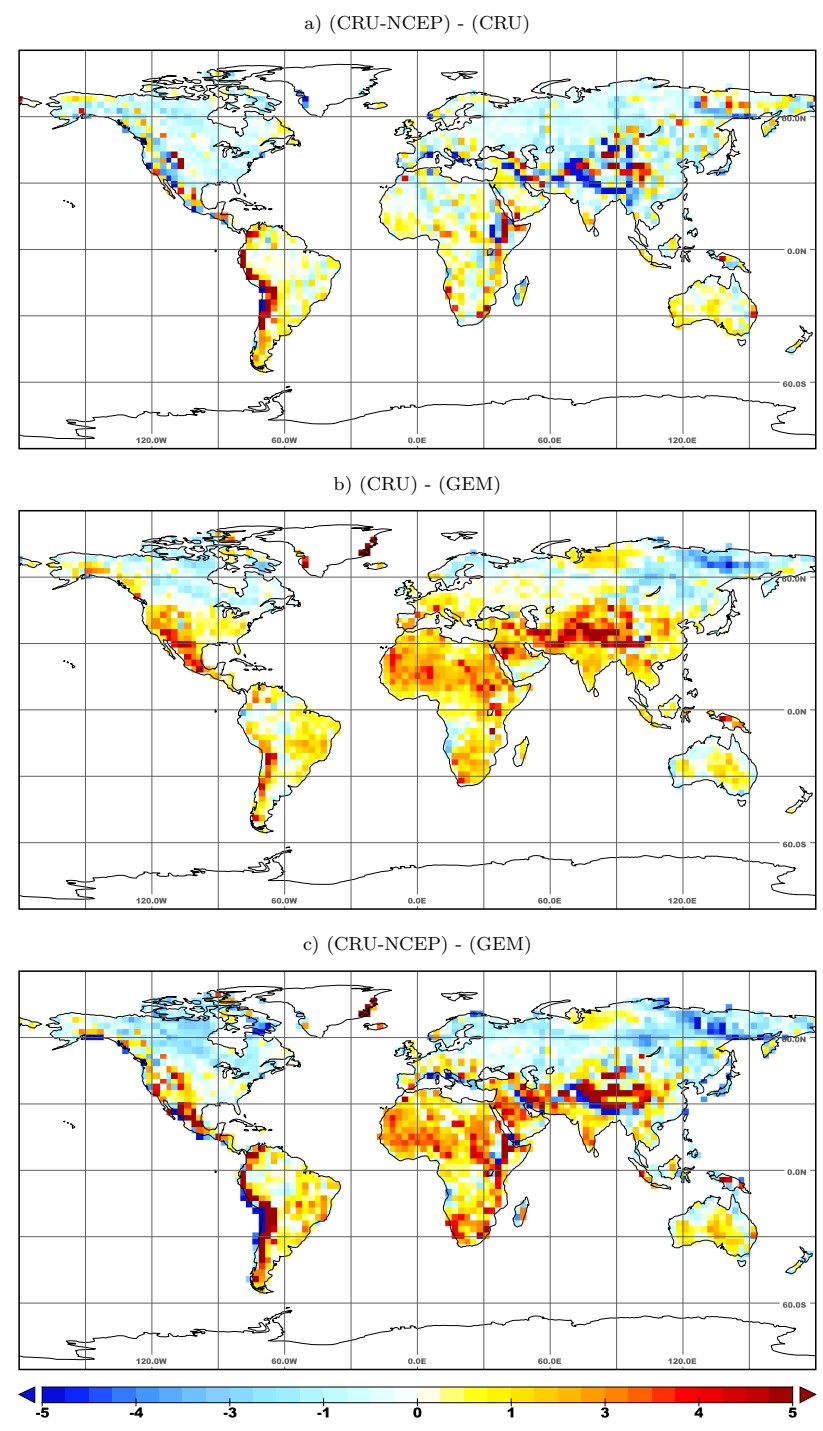

**Figure 1.** Comparison of spatial distribution patterns of annual mean temperature (°C) (averaged over the period 2009-2010): (a) CRU-NCEP minus CRU, (b) CRU minus GEM , and (c) CRU-NCEP minus GEM.

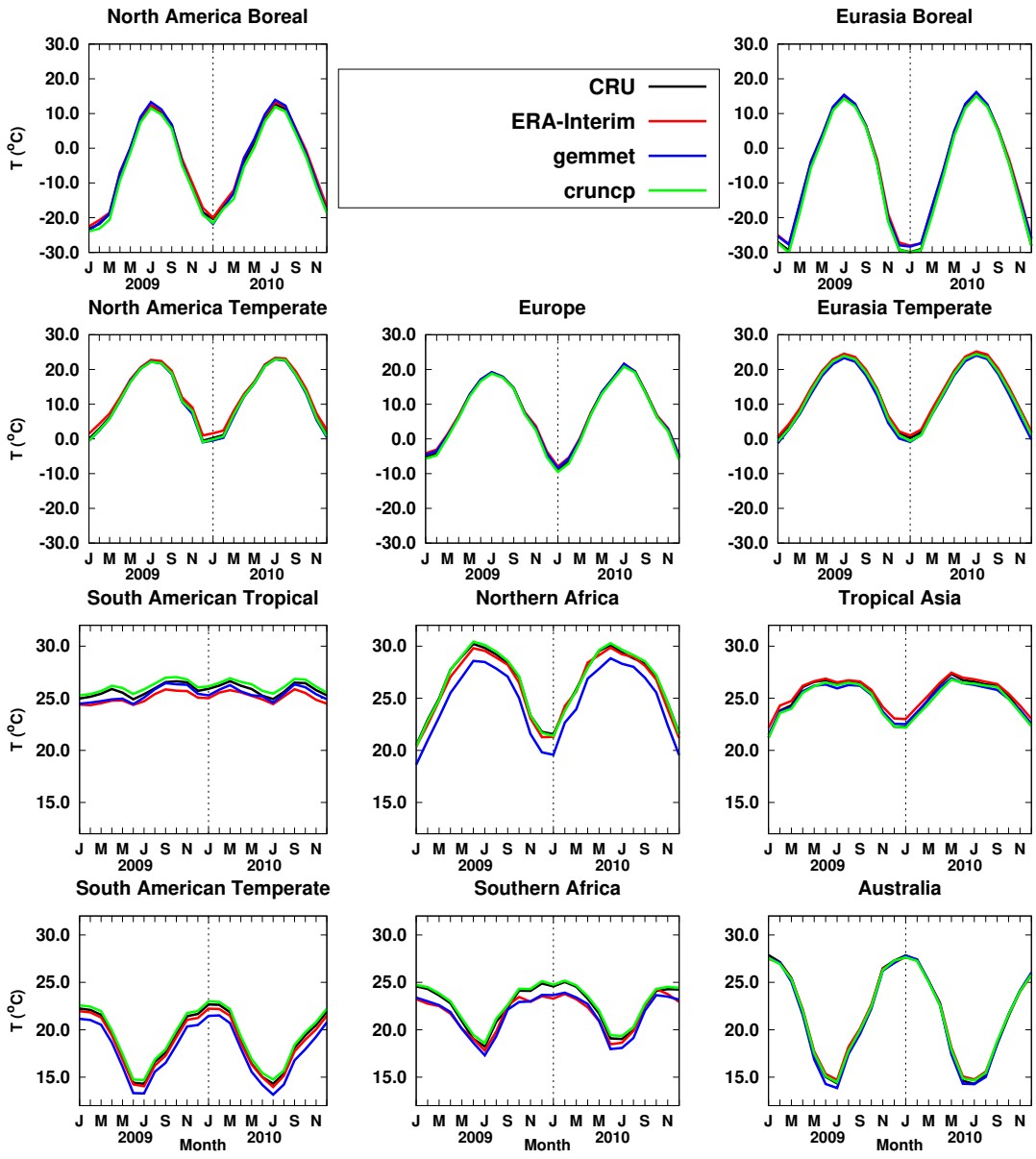

**Figure 2.** Monthly mean temperature (°C) averaged for the TransCom land regions.

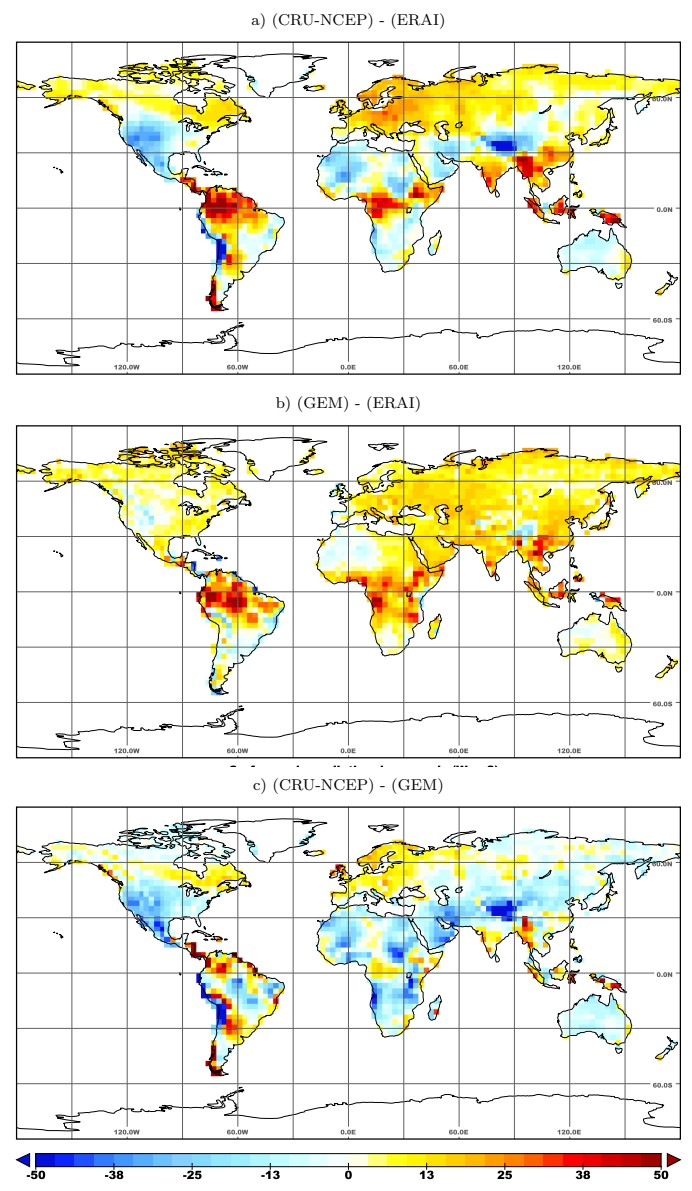

**Figure 3.** Comparison of spatial distribution patterns of annual mean shortwave radiation (W m$^{-2}$) (averaged over the period 2009-2010): (a) CRU-NCEP minus ERAI, (b) GEM minus ERAI, and (c) CRU-NCEP minus GEM.

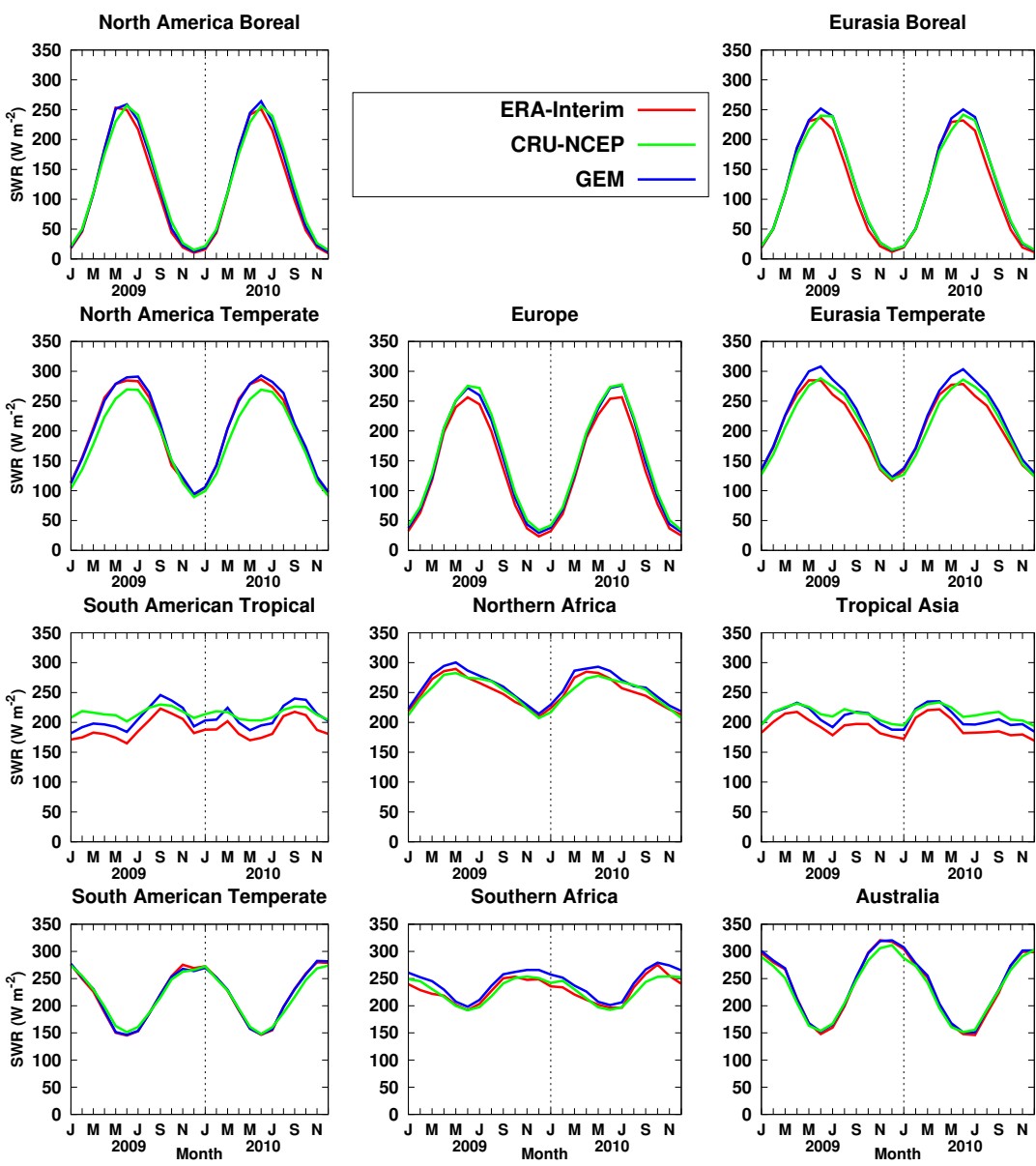

**Figure 4.** Monthly mean shortwave radiation (W m$^{-2}$) averaged for the 11 TransCom land regions.

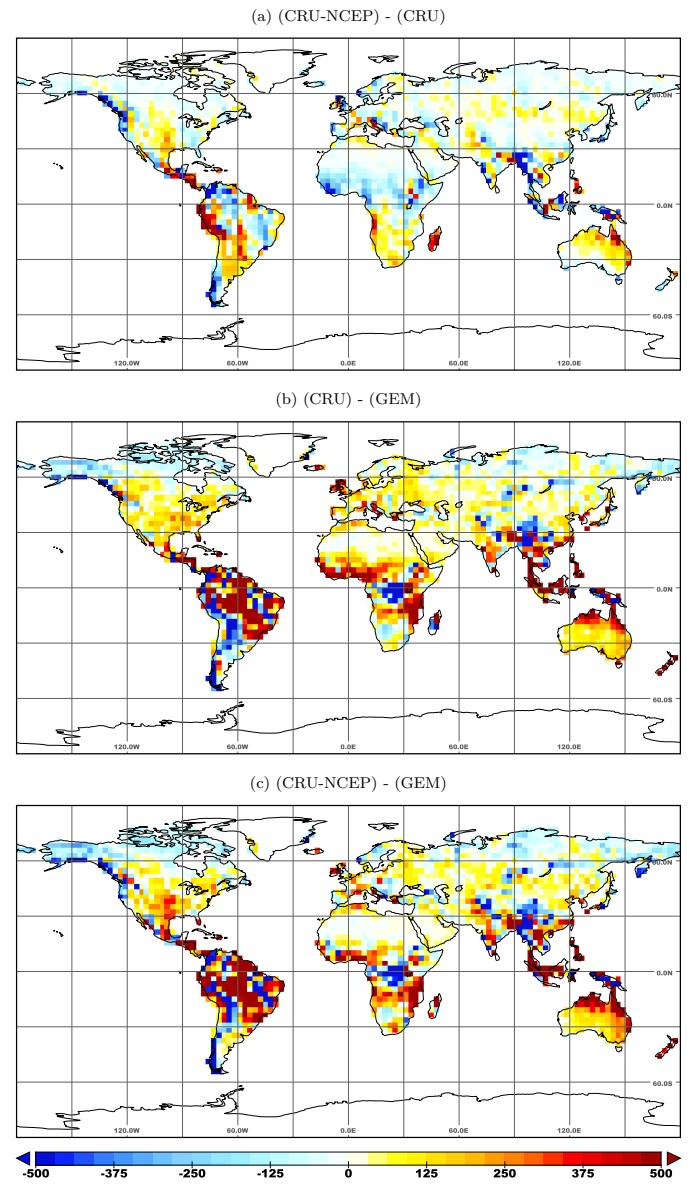

**Figure 5.** Comparison of spatial distribution patterns of annual total precipitation (mm year$^{-1}$) (averaged over the period 2009-2010): (a) CRU-NCEP minus CRU, (b) CRU minus GEM, and (c) CRU-NCEP minus GEM.

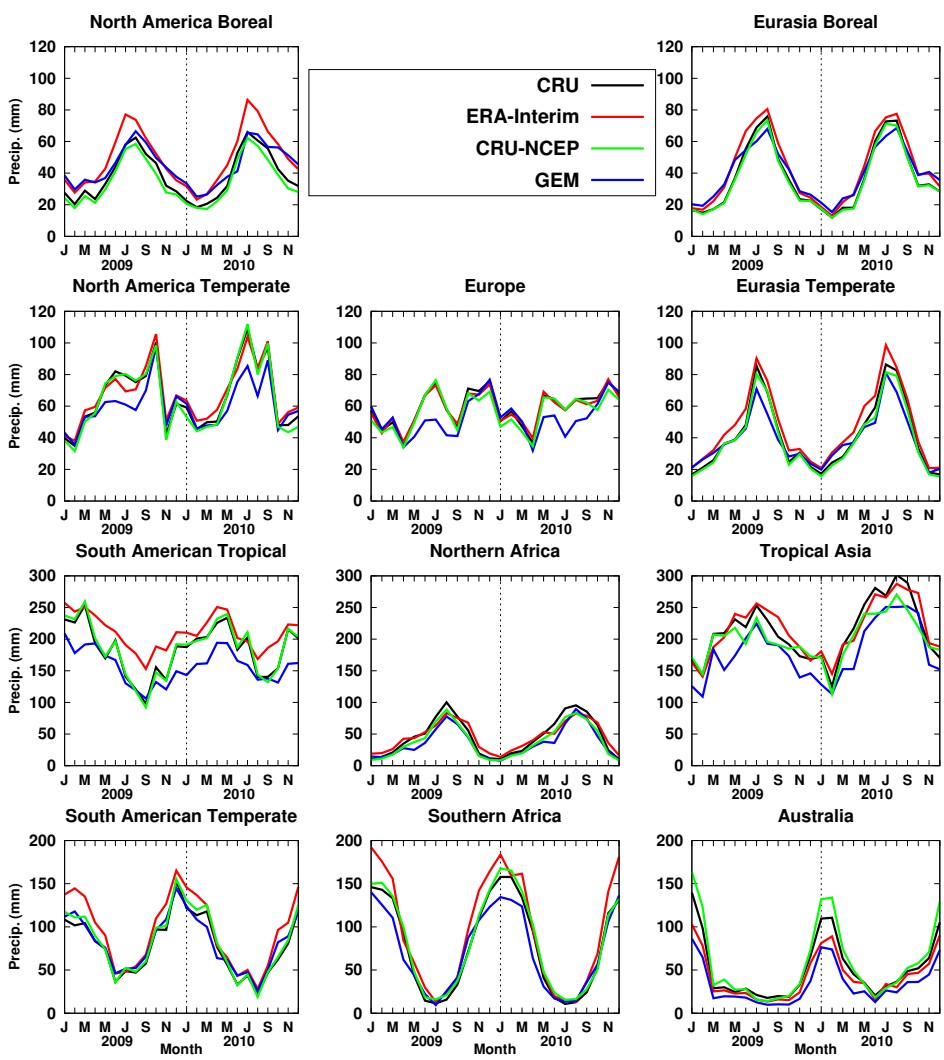

**Figure 6.** Monthly total precipitation (mm month$^{-1}$) for the 11 TransCom land regions.

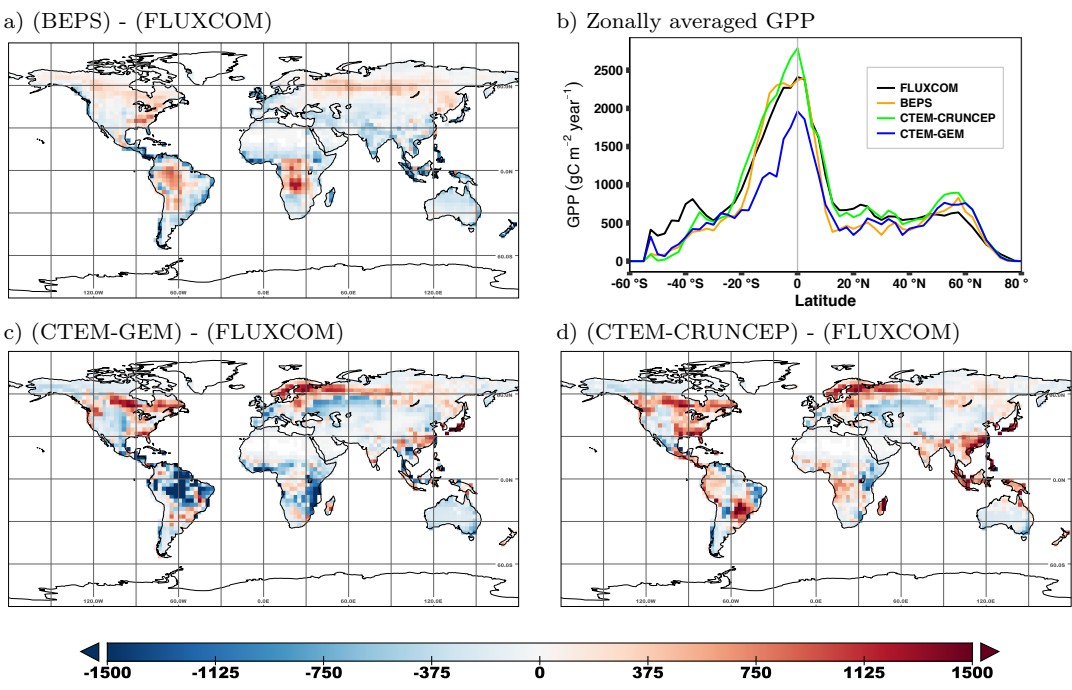

**Figure 7.** The annual spatial difference of GPP (gC m$^{-2}$ year$^{-1}$) for CTEM-GEM, CTEM-CRUNCEP, and BEPS against the observation-based GPP estimates from FLUXCOM (averaged over the period 2009-2010). The zonal distributions of GPP from all datasets are shown (top-right).

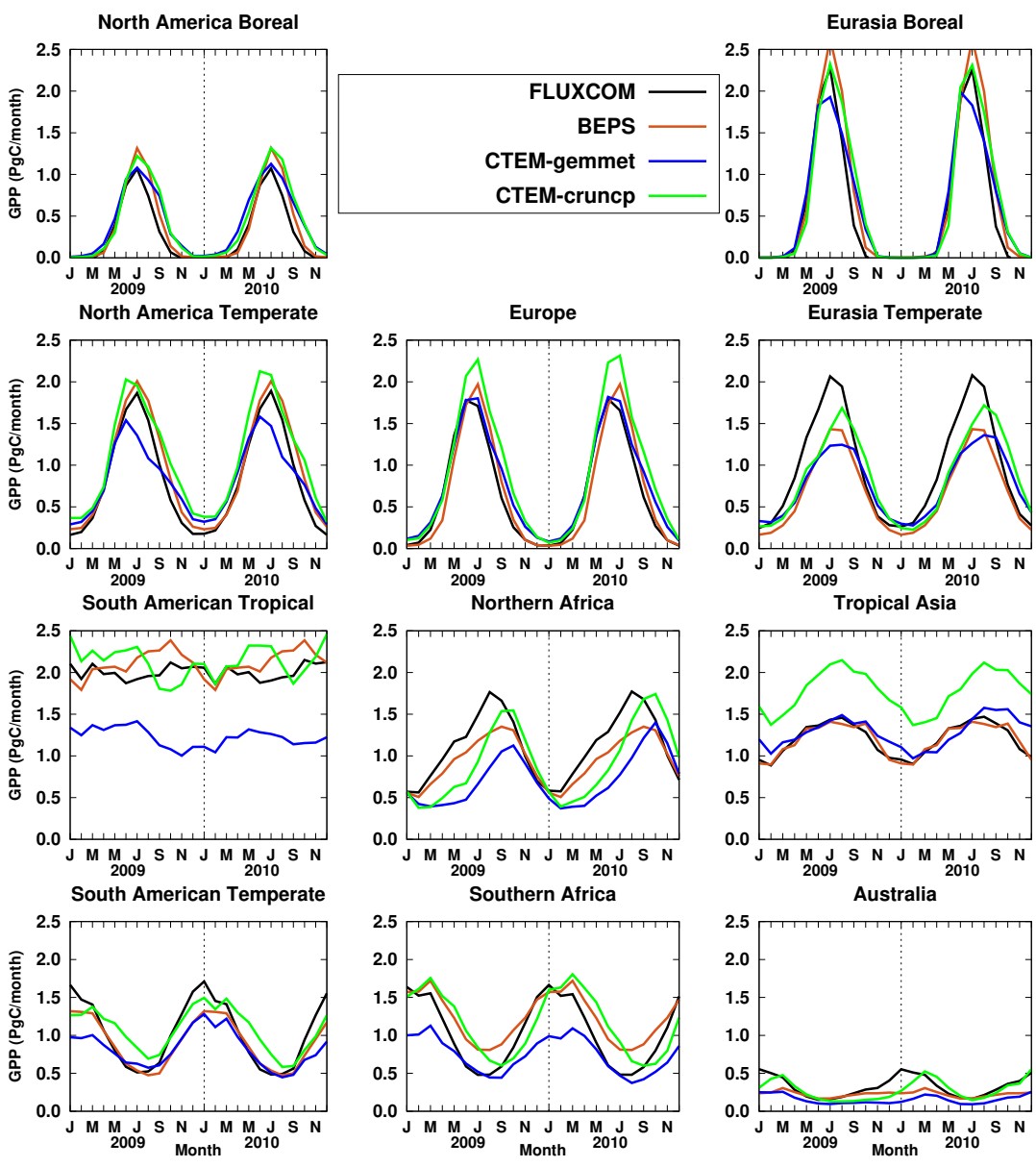

**Figure 8.** The seasonal cycle of GPP from CTEM-GEM, CTEM-CRUNCEP, BEPS, and FLUXCOM integrated over the 11 TransCom land regions.

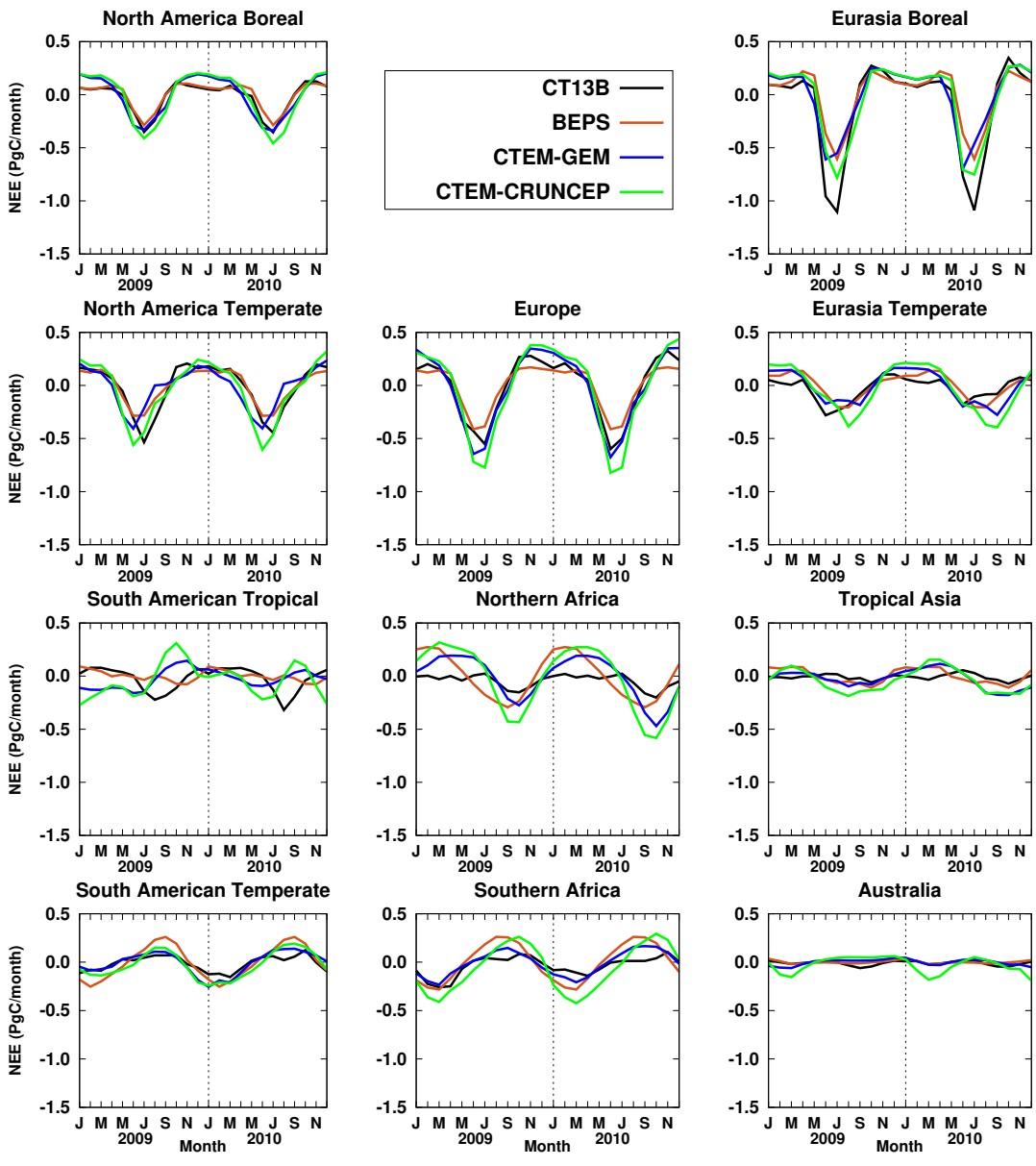

**Figure 9.** The seasonal cycle of NEE from CTEM-GEM, CTEM-CRUNCEP, and BEPS in comparison to the optimized NEE from CT2013B integrated over the 11 TransCom land regions.

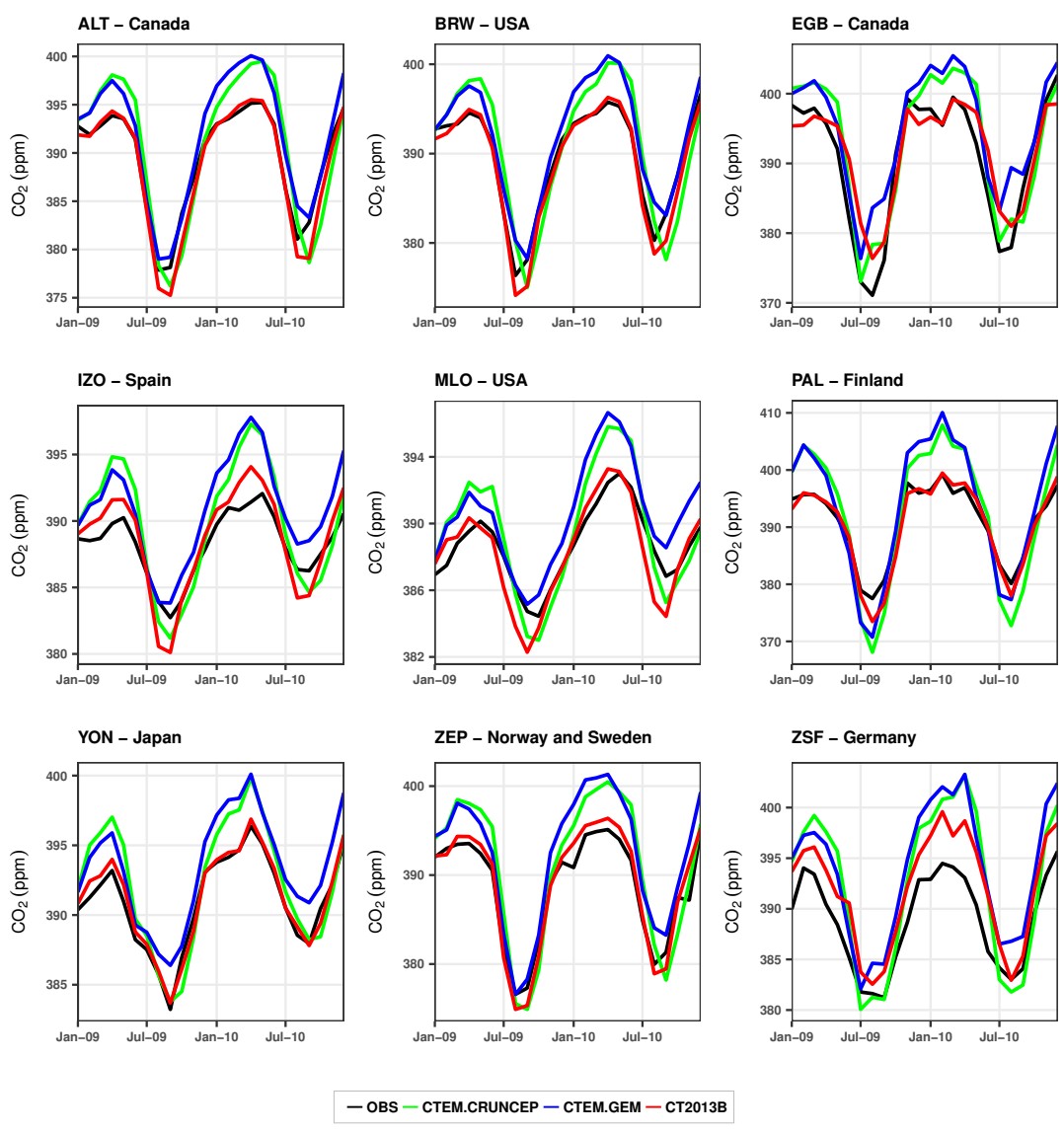

**Figure 10.** Comparison of the monthly time series of modelled $CO_2$ concentrations using land prior fluxes from CTEM-CRUNCEP (green) and CTEM-GEM (blue) with surface observations (black) at selected sites (listed in Table 1), and modelled $CO_2$ using posterior fluxes from CT2013B (red). The modelled $CO_2$ was produced by a forward run of GEM-MACH-GHG.

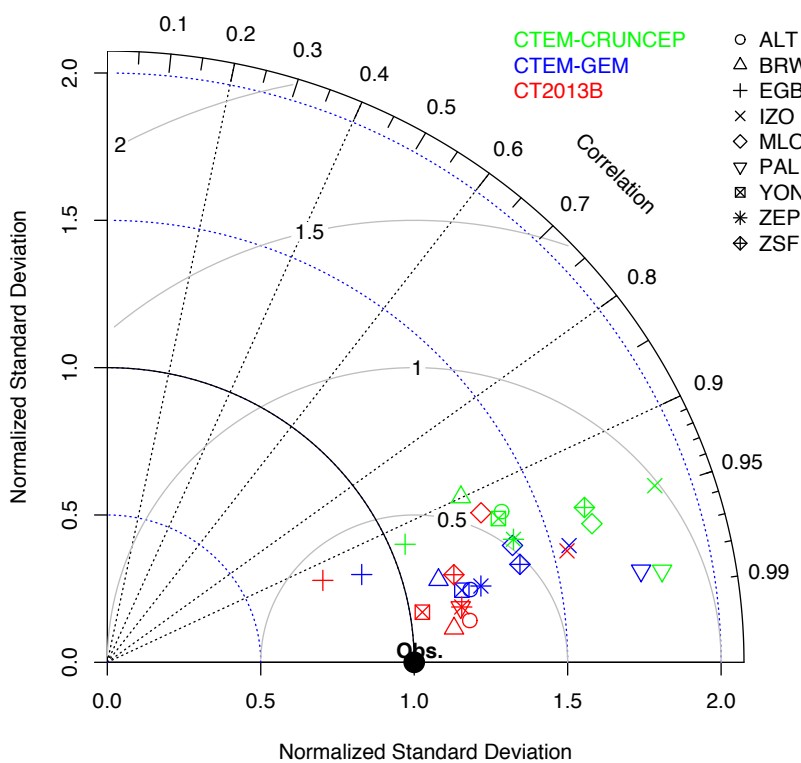

**Figure 11.** Taylor diagram of modelled monthly $CO_2$ concentrations produced by a forward run of GEM-MACH-GHG using land fluxes from CT2013B (red), CTEM-GEM (blue), and CTEM-CRUNCEP (green) and observed $CO_2$ (identified as "1.0") at selected sites (shapes) (listed in Table 1).

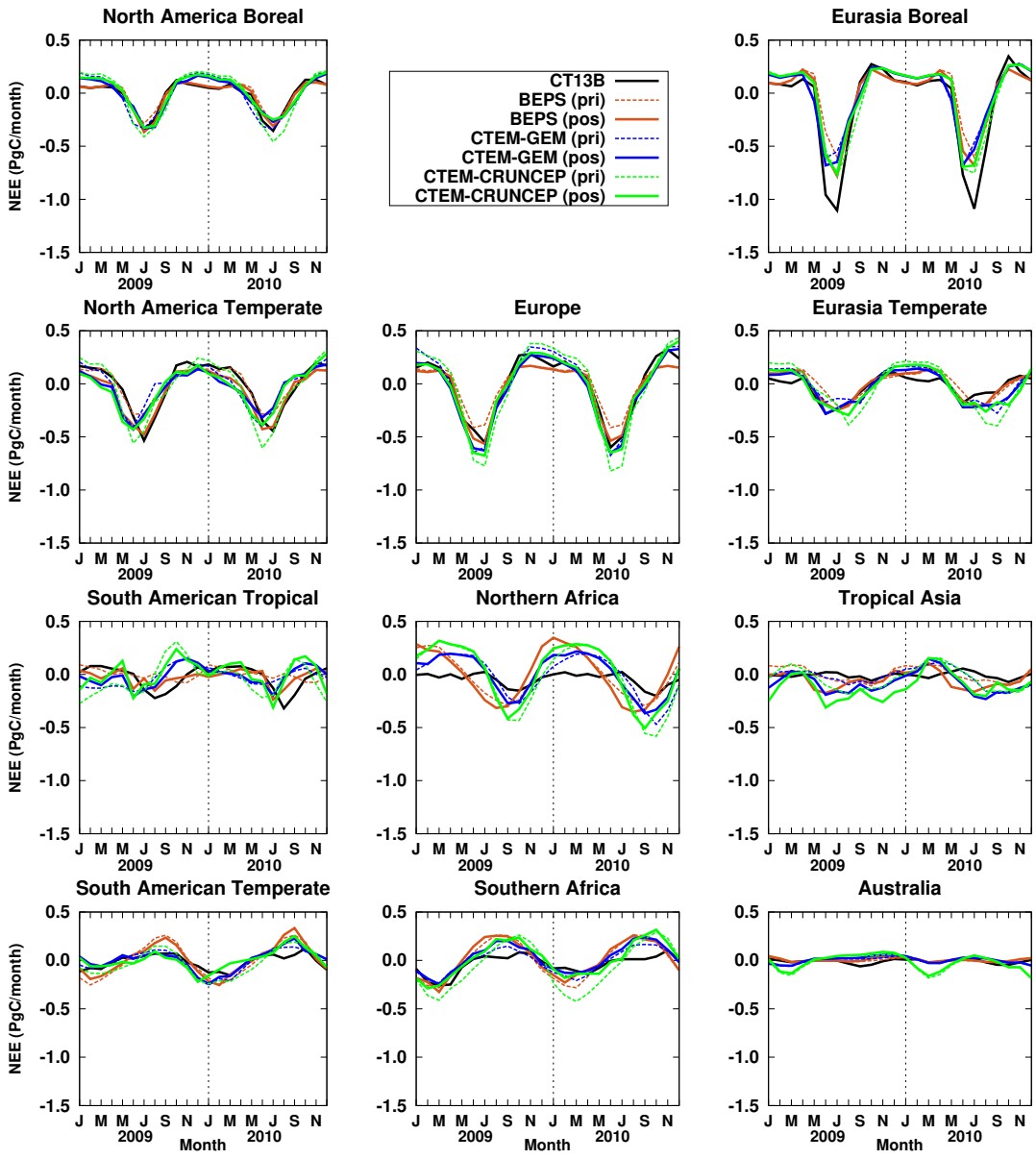

**Figure 12.** The seasonal cycle of the optimized NEE from GEOS-Chem using three different prior estimates of NEE from CTEM-GEM, CTEM-CRUNCEP, and BEPS (indicated as well) in comparison to the optimized NEE from CT2013B integrated over the 11 TransCom land regions.

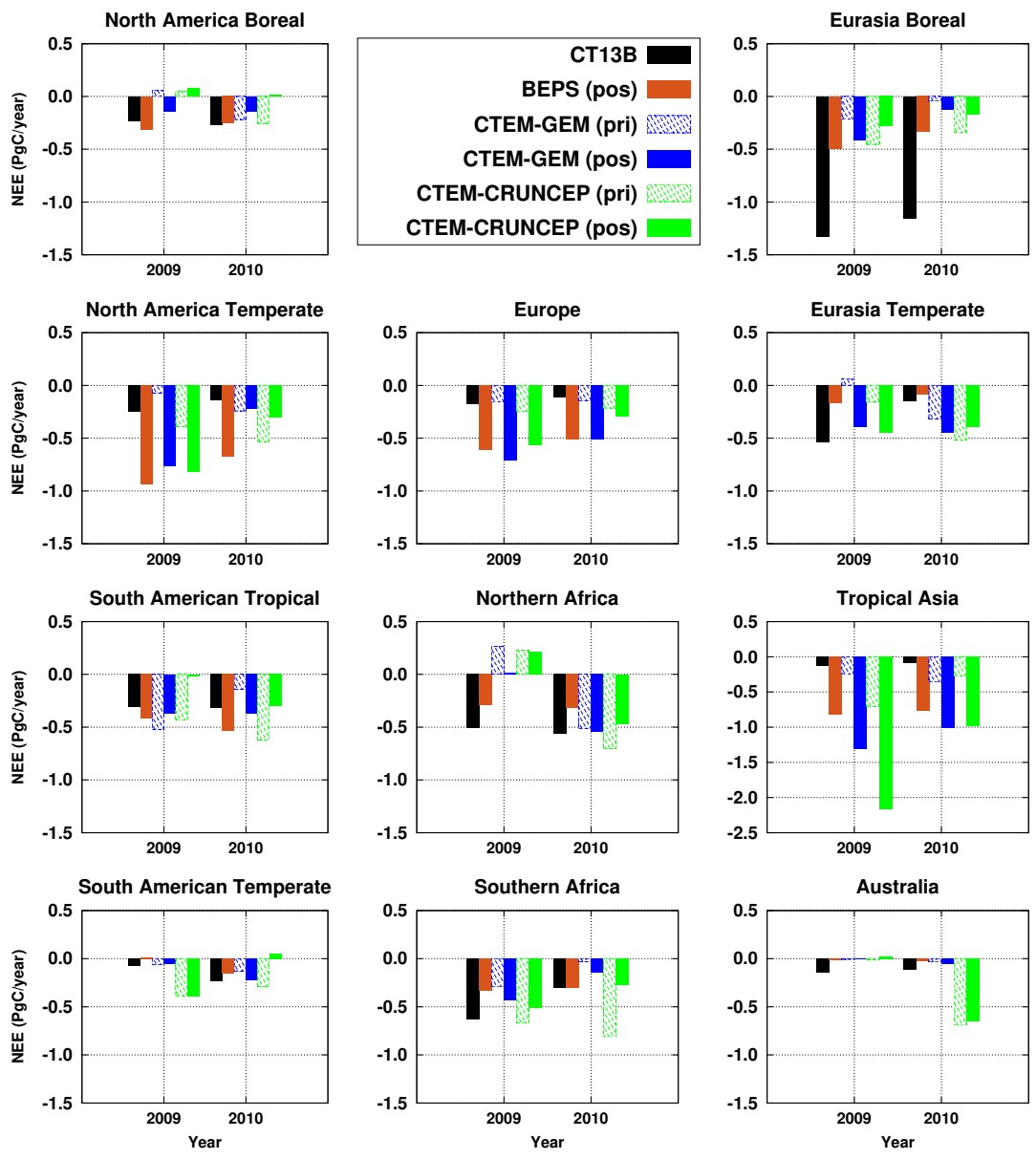

**Figure 13.** The annual total of the optimized NEE from GEOS-Chem using three different prior flux estimates of NEE from CTEM-GEM, CTEM-CRUNCEP, and BEPS (indicated as well) in comparison to the optimized NEE from CT2013B integrated over the 11 TransCom land regions.

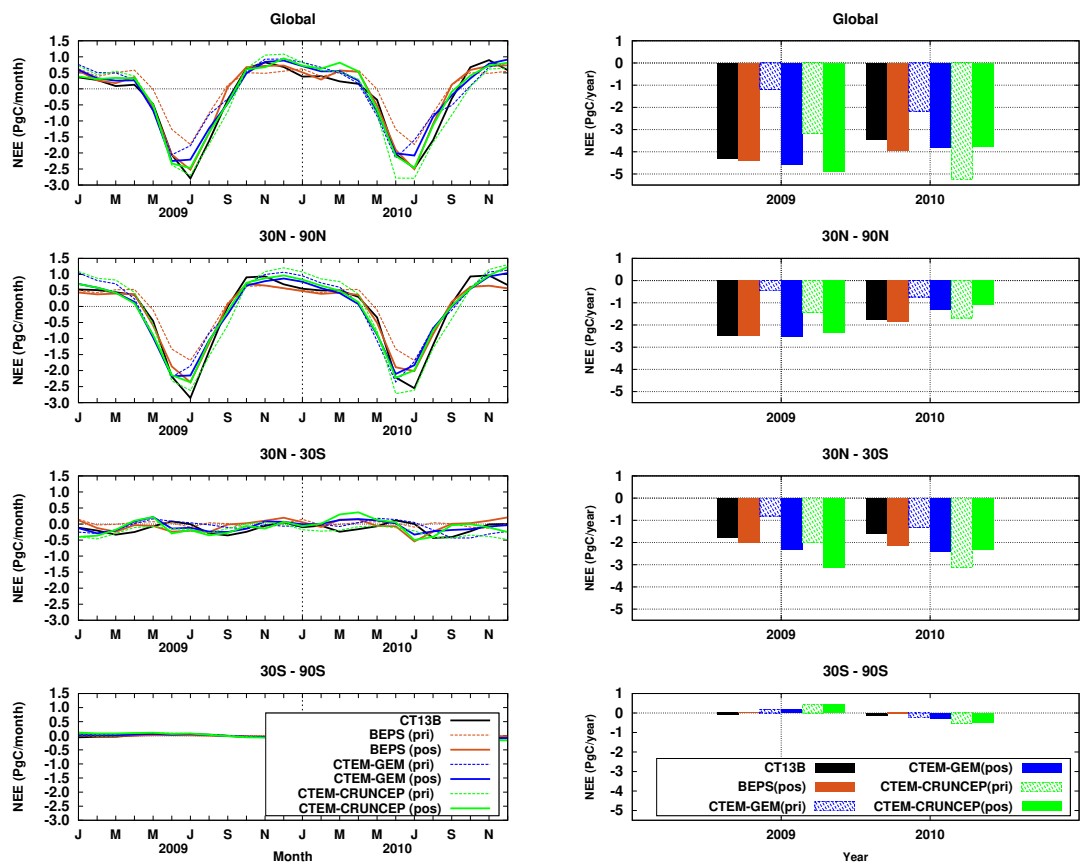

**Figure 14.** The monthly (left) and annual total (right) of the optimized NEE from GEOS-Chem using three different prior flux estimates of NEE from CTEM-GEM, CTEM-CRUNCEP, and BEPS (indicated as well) in comparison to the optimized NEE from CT2013B integrated over three latitudinal bands.

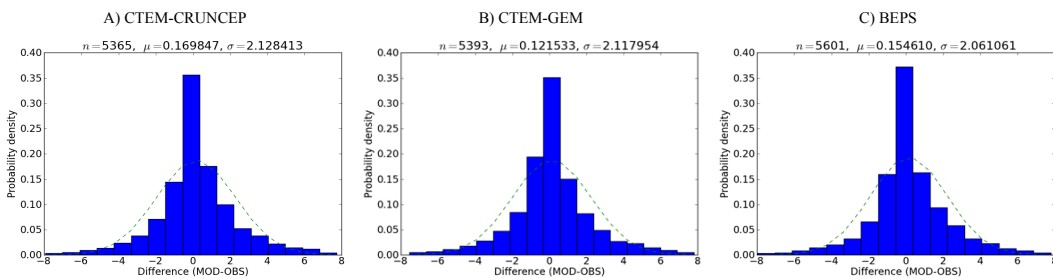

**Figure 15.** Frequency distributions of the residuals between modelled (using A) CTEM-CRUNCEP, B) CTEM-GEM, and C) BEPS prior fluxes, respectively) and observed $CO_2$ concentrations, where $n$ is the number of the assimilated observations, $\mu$ is the mean bias, and $\sigma$ is the mean standard deviation.

**Table 1.** Locations of selected stations measuring atmospheric $CO_2$ concentrations used for comparison in the forward simulations.

| Code | Site name | Country | Longitude | Latitude | Elevation (m) |
|------|-----------|---------|-----------|----------|---------------|
| ALT | Alert | Canada | -62.5 | 82.45 | 210. |
| BRW | Barrow | USA | -156.6 | 71.32 | 11. |
| EGB | Egbert | Canada | -79.8 | 44.23 | 253. |
| IZO | Tenerife | Spain | -16.5 | 28.30 | 2367. |
| MLO | Mauna Loa | USA | -155.6 | 19.54 | 3397. |
| PAL | Pallas-Sammaltunturi | Finland | 24.1 | 67.97 | 560. |
| YON | Yonagunijima | Japan | 123.0 | 24.47 | 30. |
| ZEP | Zeppelinfjelle | Norway | 11.9 | 78.90 | 475. |
| ZSF | Zugspitze | Germany | 11.0 | 47.42 | 2656. |

**Table 2.** TCCON sites used in this study.

| Site Name | Lat | Lon | Reference |
|---|---|---|---|
| Eureka, Canada | 80.05 N | 86.42 W | Strong et al. (2014) |
| Sodankyla, Finland | 67.37 N | 26.63 E | Kivi et al. (2014) |
| Bialystok, Poland | 53.23 N | 23.03 E | Deutscher et al. (2014) |
| Bremen, Germany | 53.10 N | 8.85 E | Notholt et al. (2014) |
| Karlsruhe, Germany | 49.10 N | 8.44 E | Hase et al. (2014) |
| Orleans, France | 47.97 N | 2.11 E | Warneke et al. (2014) |
| Garmisch, Germany | 47.48 N | 11.06 E | Sussmann and Rettinger (2014) |
| Park Falls, USA | 45.95 N | 90.27 W | Wennberg et al. (2014a) |
| Lamont, USA | 36.60 N | 97.49 W | Wennberg et al. (2014b) |
| Izana, Tenerife, Spain | 28.3 N | 16.5 W | Blumenstock et al. (2014) |
| Darwin, Australia | 12.42 S | 130.90 E | Griffith et al. (2014a) |
| Wollongong, Australia | 34.41 S | 150.88 E | Griffith et al. (2014b) |
| Lauder, New Zealand | 45.04 S | 169.68 E | Sherlock et al. (2014) |

**Table 3.** Simulated global values of primary carbon pools and fluxes for the spin-up simulations using CTEM-CRUNCEP, CTEM-GEM and CTEM-CRUNCEP2yr. Values are a 20 year average at the end of model simulations. Mean areal precipitation (global land and for the $30°$N-$30°$S land band) averaged for the 1901-1940 period used to spin-up CTEM-CRUNCEP, and for the 2009-2010 period used to spin-up CTEM-GEM and CTEM-CRUNCEP2yr, and the correspondence GPP estimates.

| Variable | CTEM-CRUNCEP | CTEM-GEM | CTEM-CRUNCEP2yr |
|---|---|---|---|
| Gross primary productivity (Pg C yr$^{-1}$) | 118.0 | 97.0 | 139.8 |
| Net primary productivity (Pg C yr$^{-1}$) | 58.0 | 47.0 | 70.0 |
| Autotrophic respiration (Pg C yr$^{-1}$) | 60.5 | 49.6 | 69.8 |
| Heterotrophic respiration (Pg C yr$^{-1}$) | 57.5 | 47.4 | 70.0 |
| Litter carbon respiration (Pg C yr$^{-1}$) | 40.8 | 33.4 | 49.4 |
| Soil carbon respiration (Pg C yr$^{-1}$) | 16.7 | 13.7 | 20.5 |
| Vegetation biomass (Pg C) | 674.0 | 544.0 | 829.2 |
| Litter mass (Pg C) | 97.0 | 79.0 | 108.9 |
| Soil carbon mass (Pg C) | 1410.0 | 1162.0 | 1843.0 |
| Mean areal precipitation (mm yr$^{-1}$) (global) | 760.0 | 762.0 | 828.0 |
| Mean areal precipitation (mm yr$^{-1}$) ($30°$N-$30°$S) | 1047.0 | 984.0 | 1139.0 |
| Gross primary productivity (Pg C yr$^{-1}$) ($30°$N-$30°$S) | 80.7 | 60.9 | 95.5 |

**Table 4.** Annual GPP, $R_{\mathrm{eco}}$, and NEE (PgC year$^{-1}$) from CTEM-CRUNCEP and CTEM-GEM for the transient simulations. The transient simulation was initialized from the spin-up simulations using varying $CO_2$ concentrations and meteorology.

|  | CTEM-CRUNCEP | | CTEM-GEM | | other estimates |
| --- | --- | --- | --- | --- | --- |
|  | 2009 | 2010 | 2009 | 2010 | multi-year average |
| GPP | 133.6 | 137.4 | 99.3 | 100.9 | 119.5 (Deng et al., 2014) |
|  |  |  |  |  | 123 $\pm$8 (Beer et al., 2010) |
| $R_{\mathrm{eco}}$ | 130.4 | 132.2 | 98.1 | 98.7 |  |
| NEE | -3.2 | -5.2 | -1.2 | -2.2 |  |

**Table 5.** The mean differences and RMSEs (in ppm) of the a posteriori $CO_2$ fields, based on CTEM-CRUNCEP, CTEM-GEM, and BEPS fluxes, with respect to TCCON data for July 2009 to June 2010. Station to station error is also shown.

|  | CTEM-CRUNCEP | CTEM-GEM | BEPS |
| --- | --- | --- | --- |
| Mean (mod − obs) | 0.53 | 0.64 | 0.49 |
| RMSE (mod − obs) | 1.36 | 1.36 | 1.23 |
| inter-station bias | 0.61 | 0.53 | 0.49 |

**Table 6.** The mean differences and RMSEs (in ppm) of the a posteriori $CO_2$ fields, based on CTEM-CRUNCEP, CTEM-GEM, and BEPS fluxes, with respect to aircraft data from the HIPPO-1, HIPPO-2, and HIPPO-3 campaigns.

|  |  | Mean (mod − obs) | | | RMSE (mod − obs) | | |
| --- | --- | --- | --- | --- | --- | --- | --- |
| Altitude | Latitude | CTEM-CRUNCEP | CTEM-GEM | BEPS | CTEM-CRUNCEP | CTEM-GEM | BEPS |
| 0 - 5 Km | 60°S - 30°S | 0.07 | 0.00 | 0.02 | 0.70 | 0.70 | 0.74 |
|  | 30°S - 0° | -0.61 | -0.58 | -0.33 | 0.87 | 0.83 | 0.52 |
|  | 0° - 30°N | -0.54 | -0.35 | -0.26 | 1.03 | 0.95 | 0.75 |
|  | 30°N - 60°N | -0.48 | -0.34 | -0.68 | 1.72 | 1.72 | 1.58 |
| 5 - 10 Km | 60°S - 30°S | -0.45 | -0.46 | -0.39 | 0.78 | 0.80 | 0.78 |
|  | 30°S - 0° | -0.56 | -0.44 | -0.16 | 0.80 | 0.75 | 0.69 |
|  | 0° - 30°N | -0.91 | -0.67 | -0.34 | 1.12 | 0.95 | 0.74 |
|  | 30°N - 60°N | -0.86 | -0.56 | -0.49 | 1.46 | 1.30 | 1.20 |
| 0 - 10 Km | 60°S - 60°N | -0.56 | -0.43 | -0.36 | 1.17 | 1.10 | 0.99 |