# Peer review of "Coupling the Canadian Terrestrial Ecosystem Model (CTEM v. 2.0) to Environment and Climate Change Canada's greenhouse gas forecast model (v.107-glb)"

_Geoscientific Model Development, 2017_

## Short Comment (SC1) · 5 Oct 2017

Dear authors,

in my role as Executive editor of GMD, I would like to bring to your attention our Editorial version 1.1:

http://www.geosci-model-dev.net/8/3487/2015/gmd-8-3487-2015.html

This highlights some requirements of papers published in GMD, which is also available

on the GMD website in the 'Manuscript Types' section:

http://www.geoscientific-model-development.net/submission/manuscript_types.html

In particular, please note that for your paper, the following requirements have not been met in the Discussions paper:

- "The main paper must give the model name and version number (or other unique identifier) in the title."

You applied this rule only for the first half of the title. Please include also the acronym und version number of the GHG forecast model, e.g. GEM-MACH-GHG vY.z .

Yours,

Astrid Kerkweg

---

## Author Comment (AC1) · 6 Oct 2017

Dear Executive Editor,

Thank you for your comment and the note. We will modify the title and the relevant text in the revised version of the manuscript.

The new title will be "Coupling the Canadian Terrestrial Ecosystem Model (CTEM v. 2.0) to Environment and Climate Change Canada's greenhouse gas forecast model (v.107-glb)".

[Figure]

Thank you

Bakr Badawy

---

## Referee Comment (RC1) · Anonymous Referee #1 · 14 Oct 2017

This paper describes the coupling between the Canadian Terrestrial Ecosystem Model (CTEM v2.0) and Environment and Climate Change Canada's greenhouse gas forecast model (GEM-MACH-GHG). The radiation, surface temperature, and precipitation fields etc. calculated from GEM-MACH-GHG feed to CTEMv2.0 every 30 minutes, and the net biosphere fluxes from CTEM v2.0 are used as surface boundary conditions to drive $CO_2$ simulations by GEM-MACH-GHG. The ultimate goal of this coupling is to do carbon cycle data assimilation to constrain biosphere model parameters and surface carbon fluxes. The authors tested the performance of such coupling by evaluating the

meteorology fields, and carbon fluxes including gross primary production and net terrestrial biosphere fluxes. At last, they tested the impact of using net biosphere fluxes from the coupled system as priors on atmospheric flux inversion with GEOS-Chem 4D-Var system. The paper is well structured. However, the rationale of such coupling is not well described, the evaluation of the model performance can be further improved, and some conclusions in the paper are not supported by the results. My detailed comments are below.

1. It is not clear to me why it is necessary to couple CTEM v2.0 with GEM-MACH-GHG for the purpose of carbon cycle data assimilation. CTEM v2.0 and GEM-MACH-GHG can run in parallel, and the GEM-MACH-GHG read in the fluxes from CTEM-v2.0 every 30 minutes. In that case, CTEM v2.0 can use the best possible meteorology fields it can get. As shown in this paper, the CTEM v2.0 forced by CRUNCEP performs better. I don't see the benefits of having consistent meteorology between CTEM v2.0 and GEM-MACH-GHG. The errors in meteorology affect CTEM v2.0 and GEM-MACH-GHG in quite different ways. The rational discussed in the introduction is not convincing. I would recommend adding more discussions about the benefits of coupling these two models together. If the authors can give a specific example, it would be clearer.

2. It is not clear whether the energy fluxes (e.g., latent heat flux) and water fluxes (e.g., evaporation) required running GEM-MACH-GHG is from the CTEM v2.0 or from somewhere else. I would recommend adding descriptions whether the coupling is one-way or two-way.

3. The left panels and right panels in Figure 1 are basically the same. I would recommend either plotting only one year or averaging over two years. The same applies for Figures 3, 5, and 7.

4. As discussed in the paper, precipitation from reanalysis product is not the best product available. I would recommend using Global Precipitation Climatology Project (GPCP) or CPC Merged Analysis of Precipitation (CMAP) precipitation as validation
data set.

5. In Figure 7, the authors compared model simulated GPP to an up-scaled GPP product based on flux towers (B10). But the B10 data are over different periods. The FLUXCOM that is from the same research group as B10 has GPP product over 2009 and 2010. I would recommend comparing the model simulated GPP to the FLUXCOM GPP over the same time period.

6. Figures 10 and 11 are not very informative. The differences shown in Figure 10 are a convolution of transport errors and the errors in the surface fluxes. Figure 11 uses the same transport model, so the differences are only due to surface fluxes, which have been discussed in Figure 9.

7. Tables 4 and 5 list the RMS and bias statistics of posterior $CO_2$ relative to independent $CO_2$ observations from TCCON and HIPPO campaigns. I would recommend adding one plot showing time series comparison between model simulated $CO_2$ and TCCON, and one latitudinal plot showing the comparison between model simulated $CO_2$ and HIPPO data, which may be more informative than the final statistics. I would also recommend adding a figure showing the comparison between posterior $CO_2$ and the $CO_2$ flask data assimilated in the flux inversion system, which will show how well the inversion system fits the assimilated data.

8. The inversions use the 3D $CO_2$ fields from CarbonTracker as initial $CO_2$ fields in the inversion. This is risky since the transport models between TM5 used in the CarbonTracker and GEOS-Chem is different. The initial fields works for the CarbonTracker does not necessarily the best for GEOS-Chem.

9. Some descriptions in the paper are not well justified by the results. For example, in the last paragraph in section 4.2.1: "CTEM-GEM flux estimates are within the range of the other estimates from TEMs used as a priori estimates in flux inversions (i.e., BEPS) or measurement-constrained fluxes (i.e., CT2013 B).". This is not well justified since CTEM-GEM apparently has large differences with other fluxes over the tropics due to

the bias in precipitation. This further reinforces my first point that it is not necessary to have consistent meteorology between CTEM and GEM to do carbon cycle data assimilation. It is important to have best meteorology.

---

## Referee Comment (RC2) · Anonymous Referee #2 · 16 Oct 2017

**General comments**

This paper addresses the challenge of coupling a terrestrial ecosystem model to an NWP model that has been adapted to forecast CO2 for the purpose of estimating CO2 fluxes using a flux inversion analysis system. This coupling is required in order to provide boundary conditions to the forward CO2 model and prior estimates for the flux inversion system. The main challenge stems from the differences in timescales required for the TEM to spin up and the short timescales used in NWP. The paper

presents an alternative configuration for the spin up suitable for the NWP model and compares the results with the standard spin up procedure. The results emphasize the large impact of meteorological biases on the biogenic CO2 flux biases, in particular over the tropics. Although there is a small impact on the estimation of CO2 growth rate when these fluxes are used as priors in flux inversion systems, there is a significant impact on the spatial distribution of the optimized fluxes, particularly in the tropics. All these aspects addressed are relevant scientific modelling questions within the scope of GMD, which are important to advance the use of earth system models to monitor the climate change.

The paper is well written and well structured. The methods used are valid and clearly outlined and the results are based on sound simulations with independent validation based on observations. However, the validation could be expanded to make better use of observations in the identification of regional biases both for the forward model and the optimized fluxes (see specific comments below). Test of statistical signficance would also be highly recommended in order to strengthen the conclusions drawn from the results.

**Specific Comments**

- Page 2, line 31: The sentence "Theoretically, in CCDAS, parameters of a TEM can also be optimized..." is a bit confusing. Isn't that what CCDAS aims to do?

- Page 3, line 2: The limitations of CCDAS should also be mentioned (e.g. inability to correct for model structure and missing processes).

- Page 3, line 9: Shouldn't "reduced CO2" be "increased CO2"?

- Page 6, line 15: Please remove sentence "(e.g. Agusti-Panareda et al. (2016) also had similar issues)". The TEM used in Agusti-Panareda et al. (2016) doesn't

have carbon stocks, so it does not require a spin up period.

- Page 7, line 29: Note that the TCCON observations and HIPPO data in free troposphere are not ideal to assess impact of fluxes because their sensitivity to the surface fluxes is small compared to in situ surface stations. Why weren't surface stations used for validation?

- Page 12, line 19: "the range of the other *model* estimat (Melton and Arora, 2014, Table 2)".

- Page 15, line 2: Why are only 2 sites used in the evaluation of the forward model? This evaluation is not enough to draw conclusions on the impact of the coupled fluxes on the atmospheric CO2 spatial/temporal variability at global scale. Please consider using more sites, if possible one site per TransCom region.

- Page 15, line 22: I do not agree with this statement. The differences between red and blue lines can be substantial in summer and autumn as shown in Figure 10. Please also consider the use of more sites for the validation to make the results more robust in terms of spatial distribution (see previous comment).

- Page 15, line 23-24: Again, I don't think one can say that the forward runs with the CTEM and the posterior fluxes are similar in Figure 10 when the different are around 5 ppm both at both Alert and Mauna Loa during spring, summer and autumn.

- Page 16, lines 10,11: There are significant differences in CTEM-based posterior estimates both in phase and amplites for NAmerica, Europe and Asia.

- Page 16, line 24: A small flux increment does not necessarily mean a more accurate posterior estimate. In order to conclude that it is necessary to compare

the resulting posterior estimate with independent observations of fluxes or compare posterior atmospheric CO2 concentrations with independent observations at sites that are sensitive to the biogenic fluxes.

- Page 17, line 26: Are these small differences in RMSE statistically significant?

- Page 18, lines 1,2 : These statement should be supported with a significance test of the error differences.

- Page 18, line 26: ".. datasets of analyses are simply not possible to obtain" unless re-analysis datasets are used.

- Page 19, lines 28-30: It is still not clear if the small differences in the error resulting from using CTEM-GEM or CTEM-CRUNCEP are statistically signficant. Unless this is shown, this statatement should not be used. Also, there is not explanation as to why one would expect the CTEM-GEM to provide a better constraint than CTEM-CRUNCEP for the inversion system.

- Page 19, line 31: This study provides insight into the deficiencies attributed to errors in the meteorological forcing (e.g. dry bias in precipitation over the tropics). It is not clear where is the insight into the deficiencies in the model.

- Page 19, line 33: It is not clear how can the approach in this paper help improve the performance of CLASS-CTEM. Please provide an example.

- Page 21, line 6: Please update the reference.

- Figure 8 shows a larger impact from meteorological forcing than from model formulation in NH summer, tropics and SH. This message should be emphasized in the paper.

- Figure 13 shows a large difference of NEE budget in Europe between CTEM-GEM and CTEM-CRUNCEP. This is not mentioned in the paper.

Interactive
comment

- Table 4: Please show station to station error (e.g. std of station bias) in order to evaluate the spatial variability of posterior CO2. This is commonly done in evaluation of CO2 satellite data. The ability of reproducing the global mean does not reflect the impact that the prior has on the posterior regional patterns.

- Table 5: The error with respect to the HIPPO data could be stratified in latitude bands in order to evaluate the interhemispheric gradient.
* * *

---

## Author Response (AR1)

Original comments are in black text. Our responses are in blue text.

**Associate Editor (Remarks to Author):**

In my role as Executive editor of GMD, I would like to bring to your attention our Editorial version 1.1:
http://www.geosci-model-dev.net/8/3487/2015/gmd-8-3487-2015.html
This highlights some requirements of papers published in GMD, which is also available on the GMD website in the 'Manuscript Types' section:
http://www.geoscientific-model-development.net/submission/manuscript_types.html
In particular, please note that for your paper, the following requirements have not been met in the Discussions paper:
• "The main paper must give the model name and version number (or other unique identifier) in the title."
You applied this rule only for the first half of the title. Please include also the acronym und version number of the GHG forecast model, e.g. GEM-MACH-GHG vY.z .

Thank you for your comment. We modified the title and the relevant text in the revised version of the manuscript. The new title is "*Coupling the Canadian Terrestrial Ecosystem Model (CTEM v. 2.0) to Environment and Climate Change Canada's greenhouse gas forecast model (v.107-glb)*".

**Anonymous Referee #1**

**General comments**

This paper describes the coupling between the Canadian Terrestrial Ecosystem Model (CTEM v2.0) and Environment and Climate Change Canada's greenhouse gas forecast model (GEM-MACH-GHG). The radiation, surface temperature, and precipitation fields etc. calculated from GEM-MACH-GHG feed to CTEMv2.0 every 30 minutes, and the net biosphere fluxes from CTEM v2.0 are used as surface boundary conditions to drive $CO_2$ simulations by GEM-MACH-GHG. The ultimate goal of this coupling is to do carbon cycle data assimilation to constrain biosphere model parameters and surface carbon fluxes. The authors tested the performance of such coupling by evaluating the meteorology fields, and carbon fluxes including gross primary production and net terrestrial biosphere fluxes. At last, they tested the impact of using net biosphere fluxes from the coupled system as priors on atmospheric flux inversion with GEOS-Chem 4D-Var system. The paper is well structured. However, the rationale of such coupling is not well described, the evaluation of the model performance can be further improved, and some conclusions in the paper are not supported by the results. My detailed comments are below.

We thank the referee for the careful review of our paper, and for the helpful comments. In the revised version of the paper, we significantly improve the rationale for coupling CTEM and GEM-MACH-GHG (see the response to the detailed comment #1). We believe that the revised

manuscript has greatly benefitted from the comments of both Reviewers. Below are detailed responses to each comment.

**Detailed comments**

1. It is not clear to me why it is necessary to couple CTEM v2.0 with GEM-MACH-GHG for the purpose of carbon cycle data assimilation. CTEM v2.0 and GEM-MACH-GHG can run in parallel, and the GEM-MACH-GHG read in the fluxes from CTEM-v2.0 every 30 minutes. In that case, CTEM v2.0 can use the best possible meteorology fields it can get. As shown in this paper, the CTEM v2.0 forced by CRUNCEP performs better. I don't see the benefits of having consistent meteorology between CTEM v2.0 and GEM-MACH-GHG. The errors in meteorology affect CTEM v2.0 and GEM-MACH-GHG in quite different ways. The rational discussed in the introduction is not convincing. I would recommend adding more discussions about the benefits of coupling these two models together. If the authors can give a specific example, it would be clearer.

   We agree with the reviewer that much can be accomplished without coupling CTEM v2.0 with the GEM-MACH-GHG meteorology. Specifically, prescribed fluxes can be used for flux inversions. However, there are two important reasons why eventually this coupling must be done. First, we know that theoretically it is the same meteorology that drives both the weather and ecosystem models. Thus, climate models that simulate the carbon cycle include interactive ocean and terrestrial ecosystem models. Weather forecast models are also increasingly coupled atmosphere-land-ocean models. The carbon cycle involves coupled systems as well, and eventually, coupled models may be used for greenhouse gas simulations. Second and more importantly, we are developing an ensemble Kalman Filter (EnKF) for greenhouse gas state and flux estimation. The EnKF requires realizations of prior flux errors (among other things) to directly simulate sources of forecast uncertainty. By coupling CTEM 2.0 online with the atmospheric model, we can directly perturb uncertain parameters to get an ensemble of prior fluxes. In turn, the assimilation process will be able to provide more direct feedback to the CTEM developers. Without the data assimilation prospect, CTEM is usually evaluated against flux towers at selected sites. It is because of this constant feedback on CTEM that becomes possible with data assimilation that ECCC has chosen the approach of online coupling CTEM and GEM for eventual implementation. Thus, this work helps to pave the way for the coupled meteorological and ecosystem model within the EnKF. This rationale was not explained in the original manuscript so we have added it to the introduction in the revised version.

   We have also added more discussion in the introduction about the importance of coupling bottom-up and weather models. For example, previous studies Lin et al., (2011) and Garnaud et al., (2014) (and many other, including the current paper) have shown that uncertainties in meteorological forcings (i.e. temperature, specific humidity, shortwave radiation) of ecosystem models contribute significantly to uncertainties in the simulated biospheric fluxes. At the same time, Miller et al., (2015) have shown that many of these meteorological variables (i.e. temperature, specific humidity, zonal wind, and planetary boundary layer) are also correlated with and contribute to biases in modeled atmospheric

transport. Therefore, coupling between TEMs and atmospheric transport models that account for meteorological uncertainty is considered an important step toward understanding how meteorological uncertainties impact both terrestrial $CO_2$ flux and transport errors.

Finally, we disagree that CTEM forced by CRU-NCEP performs better. A new figure (Taylor diagram in Figure 11) more clearly reveals that overall, CTEM forced by GEM generally better matches observations compared to CTEM forced by CRU-NCEP. However, the latter is indeed better in the tropics but that is where all prior (and posterior flux estimates) disagree and where observations are also inconsistent. These results can be explained by the fact that

1. The Northern Hemisphere, which is well observed, dominates the global seasonal cycle of carbon fluxes (GPP, Reco, and NEE) because it has the largest land areas that are mainly dominated by forest ecosystems. At the same time, the net contributions from the tropical and the Sothern Hemisphere regions are close to zero due to their small and opposite seasonal cycles (Figures 8 and 9).
2. Figure 9 shows that even though there is a large difference in the amplitude of the seasonal cycle of GPP from CTEM-GEM compared to CTEM-CRUNCEP, the difference is much smaller in NEE, which is what is used in the inversion. Even though GPP have large biases compared to observation-based estimates, the biases in NEE are much smaller because NEE is the difference between two large terms (GPP and Reco).

So, given that CLASS-CTEM will provide only a prior estimate of NEE (at least in the first stage) for flux inversions in EC-CAS, we feel that CTEM forced by GEM is acceptable as a source of a priori fluxes for flux inversions.

2. It is not clear whether the energy fluxes (e.g., latent heat flux) and water fluxes (e.g., evaporation) required running GEM-MACH-GHG is from the CTEM v2.0 or from somewhere else. I would recommend adding descriptions whether the coupling is one-way or two-way.

This implementation is an offline coupling in that GEM is run first and the required meteorological variables are written to an auxiliary file at half-hourly time interval. These meteorological variables are then used as atmospheric forcing for CLASS-CTEM to simulate the land carbon fluxes which then used as a priori fluxes in GEM model. This is now stated explicitly (page 9, line 2-6) as well as in the introduction (page 3, lines 33-34).

3. The left panels and right panels in Figure 1 are basically the same. I would recommend either plotting only one year or averaging over two years. The same applies for Figures 3, 5, and 7.

In the revised version of the paper, we limited the plotting to the average of 2009-2010 for Figures 1, 3, 5 and 7 as suggested by the reviewer.

4. As discussed in the paper, precipitation from reanalysis product is not the best product available. I would recommend using Global Precipitation Climatology Project (GPCP) or CPC Merged Analysis of Precipitation (CMAP) precipitation as validation data set.

First, the main focus of this comparison is to assess how the GEM meteorology differs from CRU-NCEP since that is the default forcing that is used in CLASS-CTEM. This was needed to interpret the CTEM fluxes produced with the two different sources of meteorology. To assess these differences we compared to observation-based data from CRU. We agree that GPCP and CMAP could also have been used instead of CRU but any precipitation product has its own limitations (e.g. GPCP and CMAP are based on a combination of satellite and ground observations which are affected by issues of coverage, accuracy, etc.). In particular, the different observational datasets show the largest differences in the tropics (as noted in the original manuscript (revised manuscript page 12 and 13, lines 33-35, and 1-5, respectively)). It is therefore very challenging to justify which global precipitation product is better (particularly over the tropics) to use for comparison. Second, the point of comparing to ERAI is only to get a sense of how well a reputable reanalysis product compares to independent observations. Since our meteorological products are only analyses they are not expected to perform as well as reanalysis products like ERAI or MERRA or JRA-55. Thus if reanalyses also have difficulty in matching observations, this provides context or bounds for the kind of agreement we can expect from our analyses. However, we do not condone the use of reanalyses in place of observational datasets for the purpose of validation. This, we believe, is the Reviewer's concern and we share this concern. In the revised manuscript, we have clarified this point in section 2.5 (page 8, lines 5-8).

5. In Figure 7, the authors compared model simulated GPP to an up-scaled GPP product based on flux towers (B10). But the B10 data are over different periods. The FLUXCOM that is from the same research group as B10 has GPP product over 2009 and 2010. I would recommend comparing the model simulated GPP to the FLUXCOM GPP over the same time period.

As suggested by the reviewer, we replaced B10 by FLUXCOM GPP data and changed the text accordingly. However, we first compared GPP from FLUXCOM (mean of 2009-2010) and the climatological mean GPP from B10. The comparison showed that B10 and FLUXCOM are very similar in terms of the zonal mean (Figure R1) and the spatial distribution (Figure R2). Thus, the conclusions did not change by using FLUXCOM data instead of B10.

[Figure]

Figure R1: The zonal mean of GPP from FLUXCOM and B10.

[Figure]

Figure R2: The spatial difference between FLUXCOM and B10.

6. Figures 10 and 11 are not very informative. The differences shown in Figure 10 are a convolution of transport errors and the errors in the surface fluxes. Figure 11 uses the same transport model, so the differences are only due to surface fluxes, which have been discussed in Figure 9.

Actually, Figure 10 does not involve any convolution of transport errors since the same model (GEM-MACH-GHG) was used to integrate two different sets of prior fluxes (CTEM-GEM and CTEM-CRUNCEP). The CT2013B curve shown for comparison was directly taken from the mole fractions downloaded from NOAA's website. However, we agree that Figure 10 could be much more informative. Therefore, following the suggestion from the second reviewer to "consider using more sites for the comparison in

Figure 10", we changed Figure 10 to compare the time series of the modeled and observed monthly $CO_2$ concentration at multiple sites. Also, we added a Taylor diagram (which will become figure 11) for a quantitative comparison between the modeled and observed $CO_2$ at the selected sites. We removed the figure that shows the zonal mean $CO_2$ (old Figure 11) as suggested by the Reviewer.

7. Tables 4 and 5 list the RMS and bias statistics of posterior CO2 relative to independent CO2 observations from TCCON and HIPPO campaigns. I would recommend adding one plot showing time series comparison between model simulated CO2 and TCCON, and one latitudinal plot showing the comparison between model simulated CO2 and HIPPO data, which may be more informative than the final statistics. I would also recommend adding a figure showing the comparison between posterior CO2 and the CO2 flask data assimilated in the flux inversion system, which will show how well the inversion system fits the assimilated data.

To assess the performance or tuning of the assimilation system, we normally look at the chi-squared diagnostic. This is now mentioned in the revised manuscript (section 4.3.3, page 20, lines 7-8). The chi-squared values are 0.913662, 0.892746, 0.756677 after assimilating 5365, 5393, and 5601 observations using CTEM-CRUNCEP, CTEM-GEM and BEPS, respectively. We also added histograms of modelled minus observed values over all assimilated observations (revised Figure 15). The fit of the three inversions to the assimilated observations is comparable. The standard deviations are similar and the biases are all small relative to the standard deviations. This is now mentioned in the revised text (page 20, lines 12-15).

Since we now have 15 figures in the manuscript, we are reluctant to also add the time series plots. However, we have included the time series comparisons to TCCON here for the benefit of the Reviewers and to assist with the review process. Sites at 3 different latitudes were chosen for the plots and only the middle of the 2-year study period was shown for clarity and to avoid error due to spin-up and spin-down effects. The magnitude and temporal variation of the errors for the 3 experiments are quite similar at the three sites. The largest differences are apparent at Bialystok with BEPS being more dissimilar to the two CTEM-based priors.

[Figure]

Figure R3: The temporal variation of the errors (model - obs) for 3 sites at different latitudes.

Reviewer 2 also suggested a latitudinal stratification of the HIPPO comparison. In particular, a stratification of Table 5 by latitude was requested. This was done in the revised Table 5. The text referring to Table 5 was modified accordingly (page 20, lines 19-24). We also include latitudinal plots of the HIPPO comparisons below, for the benefit of the review process, as requested.

[Figure]

**Figure R4:** Comparison of modelled and HIPPO observations as a function of latitude for observations below 5 km (top panel) and for those from 5-10 km (bottom panel).

8. The inversions use the 3D CO2 fields from CarbonTracker as initial CO2 fields in the inversion. This is risky since the transport models between TM5 used in the

CarbonTracker and GEOS-Chem is different. The initial fields works for the CarbonTracker does not necessarily the best for GEOS-Chem.

The Reviewer is correct to be concerned about the choice of initial state. However, there are two different issues associated with the initial conditions. The first is the issue of the global mean bias in the initial conditions. Any mean bias in the initial state will persist throughout the 2-year simulation length. In that regard, using the optimized 3D fields provides a means of mitigating this bias. The second issue is discrepancies in the spatial structure in the initial $CO_2$ field due to differences in transport between GEOS-Chem and TM5. On time scales of about three months, any intrahemispheric discrepancies in the initial state will be removed by large-scale mixing in the model, so on the time scale of the assimilation conducted here (2 years), these differences will not be important. Interhemispheric discrepancies, however, will matter. In our case, CarbonTracker was the best choice. Compared to the GEOS-Chem initial state, the CarbonTracker initial state actually produced less bias. This may suggest that the interhemispheric gradient in TM5 is better than in GEOS-Chem. The revised manuscript now includes this rationale for the choice of initial state (page 10, lines 31-33).

9.  Some descriptions in the paper are not well justified by the results. For example, in the last paragraph in section 4.2.1: "CTEM-GEM flux estimates are within the range of the other estimates from TEMs used as a priori estimates in flux inversions (i.e., BEPS) or measurement-constrained fluxes (i.e., CT2013 B).". This is not well justified since CTEM-GEM apparently has large differences with other fluxes over the tropics due to the bias in precipitation. This further reinforces my first point that it is not necessary to have consistent meteorology between CTEM and GEM to do carbon cycle data assimilation. It is important to have best meteorology.

While we agree with the referee that we should use the best meteorological forcings, it is challenging to identify the "best meteorology" data (precipitation, in particular) for all regions since measurements are incomplete or entirely lacking in some regions (tropics for example) [see references cited in section 4.2]. It is therefore challenging to identify the best flux estimates particularly over the tropics, where all models and observation-based estimates show large differences. Not surprisingly, CTEM has issues in the tropics, just as most other models do. This study gives us more specific insights about model deficiencies that need to be considered in the EC-CAS development processes. Moreover, even though CTEM-GEM shows large differences over the tropics, the Taylor diagram [revised Figure 11] shows that the modeled $CO_2$ using CTEM-GEM is generally relatively closer to the observed $CO_2$ compared to the one from CTEM-CRUNCEP (for both the forward and inversion simulations (revised Figure 11 and Figures R5, R6 below). Also Table A2 below shows that CTEM-GEM-based $CO_2$ fields have relatively smaller RMSEs at most sites than CTEM-CRUNCEP. This further supports our view that CTEM-GEM provides reasonable a priori-fluxes that are consistent with the atmospheric $CO_2$ signal.

**Table A1**: Locations of selected tropical stations measuring atmospheric $CO_2$ concentrations used in the inversion simulations.

| Code | Name | Country | Latitude | Longitude |
|------|------|---------|----------|-----------|
| ASC | Ascension Island | United Kingdom | -7.967 | -14.400 |
| ASK | Assekrem | Algeria | 23.262 | 5.632 |
| BKT | Bukit Kototabang | Indonesia | -0.202 | 100.318 |
| EIC | Easter Island | Chile | -27.160 | -109.428 |
| IZO | Izana, Tenerife, Canary Islands | Spain | 28.309 | -16.499 |
| NMB | Gobabeb | Namibia | -23.580 | 15.030 |
| RPB | Ragged Point | Barbados | 13.165 | -59.432 |
| SEY | Mahe Island | Seychelles | -4.682 | 55.532 |

**Table A2:** The mean differences and RMSEs (in ppm) of the posterior $CO_2$ fields, based on CTEM-CRUNCEP, CTEM-GEM, and BEPS fluxes, with respect to surface observations at selected tropical sites (Table A1).

| | Mean (mod − obs) | | | RMSE (mod − obs) | | |
|------|-------------|----------|------|-------------|----------|------|
| | CTEM-CRUNCEP | CTEM-GEM | BEPS | CTEM-CRUNCEP | CTEM-GEM | BEPS |
| BKT | -0.08 | 0.86 | 0.15 | 2.70 | 2.96 | 3.14 |
| NMB | 0.34 | 0.19 | 0.30 | 1.67 | 1.24 | 1.24 |
| ACS | 0.05 | 0.12 | 0.09 | 0.97 | 0.83 | 0.85 |
| ASK | -0.10 | -0.07 | -0.03 | 0.91 | 0.72 | 0.94 |
| RPB | -0.28 | -0.23 | -0.18 | 0.91 | 0.68 | 0.91 |
| IZO | 0.41 | 0.34 | 0.39 | 1.67 | 1.41 | 1.67 |
| SEY | -0.27 | -0.16 | -0.24 | 0.91 | 0.80 | 0.84 |
| EIC2 | 0.42 | 0.49 | 0.47 | 0.93 | 0.94 | 0.95 |

[Figure]

**Figure R5.** Comparison of the posteriori modelled $CO_2$ using land the optimized fluxes from CTEM-CRUNCEP (blue circle), CTEM-GEM (blue diamond), and BEPS (green) with surface observations (red) at selected sites (Table A1). The modelled $CO_2$ produced by GEOS-Chem model.

[Figure]

**Figure R6:** Taylor diagram of modeled $CO_2$ produced by GEOS-Chem model using land fluxes from BEPS (Orange), GEM (Blue), and CRUNCEP (Green) and observed $CO_2$ (identified as "1.0") at selected sites (shapes) (listed in Table A1).

**Anonymous Referee #2**

**General comments**

This paper addresses the challenge of coupling a terrestrial ecosystem model to an NWP model that has been adapted to forecast CO2 for the purpose of estimating CO2 fluxes using a flux inversion analysis system. This coupling is required in order to provide boundary conditions to the forward CO2 model and prior estimates for the flux inversion system. The main challenge stems from the differences in timescales required for the TEM to spin up and the short timescales used in NWP. The paper

presents an alternative configuration for the spin up suitable for the NWP model and compares the results with the standard spin up procedure. The results emphasize the large impact of meteorological biases on the biogenic CO2 flux biases, in particular over the tropics. Although there is a small impact on the estimation of CO2 growth rate when these fluxes are used as priors in flux inversion systems, there is a significant impact on the spatial distribution of the optimized fluxes, particularly in the tropics. All these aspects addressed are relevant scientific modelling questions within the scope of GMD, which are important to advance the use of earth system models to monitor the climate change.

The paper is well written and well structured. The methods used are valid and clearly outlined and the results are based on sound simulations with independent validation based on observations. However, the validation could be expanded to make better use of observations in the identification of regional biases both for the forward model and the optimized fluxes (see specific comments below). Test of statistical signficance would also be highly recommended in order to strengthen the conclusions drawn from the results.

We are grateful to the referee for the careful review of our paper, and for the useful comments. We believe that the revised manuscript has greatly benefitted from the comments of both Reviewers. A point-by-point response to each comment is provided below.

**Specific Comments**

1. Page 2, line 31: The sentence "Theoretically, in CCDAS, parameters of a TEM can also be optimized..." is a bit confusing. Isn't that what CCDAS aims to do?

   In retrospect, we agree with the reviewer that the statement is confusing. Our original intention was to contrast the CCDAS approach from inversion systems (such as CarbonTracker or those based on GEOS-Chem) which optimize surface fluxes but not parameters of the ecosystem models. However, the intention was not realized. Therefore, we have changed this statement to the following "In CCDAS, key parameters of a TEM can also be optimized..."

2. Page 3, line 2: The limitations of CCDAS should also be mentioned (e.g. inability to correct for model structure and missing processes).

   This is a good point. We added one sentence (section 1, Page 3, line 4-6) regarding the limitation of CCDAS including references.

3. Page 3, line 9: Shouldn't "reduced CO2" be "increased CO2"

   Yes. We changed "reduced $CO_2$" to "reduced $CO_2$ uptake" to be more consistent with the previous sentence.

4. Page 6, line 15: Please remove sentence "(e.g. Agusti-Panareda et al. (2016) also had similar issues)". The TEM used in Agusti-Panareda et al. (2016) doesn't have carbon stocks, so it does not require a spin up period.

   The original intention was not to point out issues with the spin up period but rather the challenges of dealing with a constantly changing system. Agusti-Panareda et al. (2016) encountered this issue when trying to construct a consistent, long-term model climatology. However, the reviewer is correct that the statement as it stands implies something else. Since it is not worth trying to explain this reference, we removed the statement as suggested by the Reviewer.

5. Page 7, line 29: Note that the TCCON observations and HIPPO data in free troposphere are not ideal to assess impact of fluxes because their sensitivity to the surface fluxes is small compared to in situ surface stations. Why weren't surface stations used for validation?

   Flask observations from surface stations were assimilated by GEOS-Chem in the inversion so they only provide a check on the consistency or set-up of the assimilation system. We still need to compare to independent observations that were not assimilated for validation. That is why we chose to focus on the TCCON and HIPPO comparisons and to not show the comparisons to surface data. The reason for focusing on TCCON and HIPPO is now added to the manuscript (page 20, lines 16-18).

   Continuous surface observations are other sources of independent observations, but they provide limited insight (based on many years of experience) because of the coarse 4°×5° resolution used by GEOS-Chem inversions. Specifically, comparisons to continuous measurements reveal the ability to capture the synoptic, seasonal and longer timescales well. But this is also evident from comparisons to TCCON observations. It is not possible to capture the finer time scale variations with the coarse resolution we use. Again, this is already well known from our experience with the system. Thus, this observation set does not much help our specific goal here of studying the different prior fluxes in the context of our flux inversions.

Nevertheless, comparisons to the surface flask data that were assimilated are now better described. Chi-squared diagnostics are mentioned (page 20, lines 7-12) and histograms of the residuals between modelled and observed values are included in the revised manuscript (revised Figure 15). These comparisons serve to assess the setup of the inversion system and all 3 experiments fit the assimilated data about equally well.

6. Page 12, line 19: "the range of the other *model* estimat (Melton and Arora, 2014, Table 2)".

   Changed

7. Page 15, line 2: Why are only 2 sites used in the evaluation of the forward model? This evaluation is not enough to draw conclusions on the impact of the coupled fluxes on the atmospheric CO2 spatial/temporal variability at global scale. Please consider using more sites, if possible one site per TransCom region.

   As suggested, we have replaced Figure 10 with a multi-panel plot of time series comparisons and added a Taylor diagram comparing the modeled and observed $CO_2$ at 9 sites as a revised Figure 11 that are representative of various global regions. Also, we removed the original Figure 11 to reduce the number of figures.

8. Page 15, line 22: I do not agree with this statement. The differences between red and blue lines can be substantial in summer and autumn as shown in Figure 10. Please also consider the use of more sites for the validation to make the results more robust in terms of spatial distribution (see previous comment).

   In response to the previous point, we have replaced the original Figure 11 with a Taylor diagram comparing the modeled and observed $CO_2$ at multiple sites, and the text has been changed accordingly.

9. Page 15, line 23-24: Again, I don't think one can say that the forward runs with the CTEM and the posterior fluxes are similar in Figure 10 when the different are around 5 ppm both at both Alert and Mauna Loa during spring, summer and autumn.

   We changed the whole text in section 4.2.2 to discuss the new figures (10 and 11) showing the monthly time series and the Taylor diagram of the modeled and observed $CO_2$ at multiple sites, in response to comment #8.
   Nevertheless, we agree with the reviewer regarding the original Figure. However the new figures show that the general pattern of the seasonal cycle from CTEM simulations is fairly good in comparison to CT2013B, given that CTEM fluxes are not yet optimized. Interestingly, during the NH growing season, in particularly in 2009, CTEM-GEM is closer to the observation than both CT2013B and CTEM-CRUNCEP at ALT, BRW, IZO, MLO, and ZEP. Figure 9 shows that both CTEM-GEM and CTEM-CRUNCEP simulate higher $CO_2$ fluxes during winter compared to CT2013B, which can explain the enhanced $CO_2$ concentration during the winter season at the selected sites from the CTEM-based simulations. This also suggests that the model structure has some

deficiencies in simulating carbon fluxes during the winter season. This point has been added to the revised version (page 17, line 10-15; page 19, line 6-7; and page 22, lines 1-7).

10. Page 16, lines 10,11: There are significant differences in CTEM-based posterior estimates both in phase and amplites for NAmerica, Europe and Asia.

We totally agree with the reviewer that there are significant differences in CTEM-based posterior estimates both in phase and amplitudes in NH regions. But, what was meant here is that the spread in the NH regions is much smaller than that in the tropical regions. We had already mentioned the large difference in Eurasia boreal. We have clarified this point in the revised version. Also, we bear in mind that CT2013 is not the truth and it is not well constrained over many regions of the globe, in particular the tropics, where measurements are incomplete or entirely lacking.

11. Page 16, line 24:  A small flux increment does not necessarily mean a more accurate posterior estimate.  In order to conclude that it is necessary to compare the resulting posterior estimate with independent observations of fluxes or compare posterior atmospheric CO2 concentrations with independent observations at sites that are sensitive to the biogenic fluxes.

We meant here that the a posteriori from CTEM-CRUNCEP has been shifted to be almost close to the a priori CTEM-GEM which didn't change much. The comparison of the modeled and observed $CO_2$ for the forward simulation showed that CTEM-GEM is much closer to observation than CTEM-CRUNCEP, which might explain why the optimized fluxes from CTEM-GEM didn't change much from it's a priori value. This point has been clarified in the revised version (see page 18, lines 28-33). We also compared the optimized $CO_2$ fields with independent data from TCCON and HIPPO and the results show that CTEM-GEM have a good agreement with these observations. In addition, in a subsequent analysis (Byrne et al., 2017) we conducted a more detailed evaluated the CTEM-based a priori fluxes for NH ecosystems and found that CTEM-GEM does indeed provide a better simulation of atmospheric $CO_2$ data than CTEM-CRUNEP. The Byrne et al. (2017) manuscript is in review in the Journal of Geophysical Research, but we now cite it in the revised manuscript in sections 4.2.1 and 4.3.1.

12.  Page 17, line 26: Are these small differences in RMSE statistically significant?
We thank the reviewer for highlighting this point. The differences in the RMSEs between CTEM-CRUNCEP and CTEM-GEM are not statistically significant (added in page 20, line 28-29). Our subsequent analysis in Byrne et al. (submitted, 2017) does indicate that the a priori CTEM-GEM provide a better simulation of variability in atmospheric CO2, but the statistics in this manuscript does not support the argument that the a posteriori fields based on CTEM-GEM are better. Furthermore, for the revised manuscript we restricted the TCCON comparison period to July 2009 to June 2010  and we found that the RMSEs are the same for CTEM-CRUNCEP and CTEM-GEM (see revised Table 5).

13. Page 18, lines 1,2 : These statement should be supported with a significance test of the error differences.
    As noted above, it is not supported by the significance test so the statement was changed.

14. Page 18, line 26: ".. datasets of analyses are simply not possible to obtain" unless re-analysis datasets are used.

    We changed it as suggested.

15. Page 19, lines 28-30: It is still not clear if the small differences in the error resulting from using CTEM-GEM or CTEM-CRUNCEP are statistically signficant. Unless this is shown, this statement should not be used. Also, there is not explanation as to why one would expect the CTEM-GEM to provide a better constraint than CTEM-CRUNCEP for the inversion system.
    Actually, based on the results of Byrne et al. (submitted, 2017) we would expect CTEM-GEM to be better. However, in the context of the analyses conduced here, we do not necessarily expect CTEM-GEM to provide a better constraint than CTEM-CRUNCEP for the inversion, given the far-from-ideal spinup process used for the former. The key points for our analysis here are (1) whether deficiencies in the CTEM-GEM prior fluxes would be evident in the context of a flux inversion when observations can correct for prior flux errors, and (2) how the deficiencies in CTEM-GEM prior fluxes compare to other prior fluxes. If they are no worse than other sources of prior fluxes, then they are a potential starting point for our flux estimation system. The differences, as noted by the reviewer, are small and are not statistically significant. We have, therefore, removed the statement. Instead, we acknowledge that CTEM-GEM can provide useful prior fluxes for flux inversions.

16. Page 19, line 31: This study provides insight into the deficiencies attributed to errors in the meteorological forcing (e.g. dry bias in precipitation over the tropics). It is not clear where is the insight into the deficiencies in the model.

    Here we were referring to the deficiencies in simulating the seasonal cycle from CTEM noted in Page 19- lines 7-13. These deficiencies were uncovered in the course of our study and the authors of CTEM (Arora, Melton) ended up revising CTEM's formulation to address this issue. We tried to clarify this point in the revised version (page 17, line 10-15; page 19, line 6-7; and page 22, lines 1-7). We also mention the deficiencies in the model regarding the high $CO_2$ fluxes during the winter season (see the response to the comment #9).

17. Page 19, line 33: It is not clear how can the approach in this paper help improve the performance of CLASS-CTEM. Please provide an example.

    As mentioned in the previous comment, the analysis shows some deficiencies in simulating the phase of the seasonal cycle as well as the winter fluxes which will form the basis for future development of CLASS-CTEM. The developers of CLASS-CTEM (Arora, Melton) are co-authors in this paper so they are aware about this issue and are

presently working on solving it. However, we agree that this detail was not evident in the original manuscript. This has now been clarified in the revised version (page 17, line 10-15; page 19, line 6-7; and page 22, lines 1-7), and we revised the relevant conclusion statement.

18. Page 21, line 6: Please update the reference.

Done.

19. Figure 8 shows a larger impact from meteorological forcing than from model formulation in NH summer, tropics and SH. This message should be emphasized in the paper.

This is a very good point and we are grateful to the Reviewer for this insight. We now explicitly mention that the impact of meteorological forcings can exceed that due to model formulation. We also note that we demonstrate that using different meteorological forcings result in the same biases in the winter season, indicating there is a problem in the model phenology as well. We now mention this in different places in the paper and have added it to the conclusions as well.

20. Figure 13 shows a large difference of NEE budget in Europe between CTEM-GEM and CTEM-CRUNCEP. This is not mentioned in the paper.

We will mention to this in the revised paper (page 19, lines 19-20). But, basically there are large differences in all regions, which highlight the sensitivity of the inversion to the a priori fluxes (as noted in the paper).

21. Table 4: Please show station to station error (e.g. std of station bias) in order to evaluate the spatial variability of posterior CO2. This is commonly done in evaluation of CO2 satellite data. The ability of reproducing the global mean does not reflect the impact that the prior has on the posterior regional patterns.

Table 4 has now been revised to include station to station error and the text had been modified to mention this.

22. Table 5: The error with respect to the HIPPO data could be stratified in latitude bands in order to evaluate the interhemispheric gradient.

Table 5 was modified as suggested and the revised manuscript now discusses the interhemispheric gradient.

[revised manuscript text omitted]